psychology

familiar face identity, natural images, fast periodic visual stimulation, EEG, inversion

**Author for correspondence:**
Bruno Rossion
e-mail: bruno.rossion@univ-lorraine.fr

# An objective, sensitive and ecologically valid neural measure of rapid human individual face recognition

Friederike G. S. Zimmermann[1,2,†], Xiaoqian Yan[1,†]
and Bruno Rossion[1,3,4]

[1]Institute of Research in Psychological Science, Institute of Neuroscience, Université de Louvain, Louvain-la-Neuve, Belgium
[2]BG Klinikum Hamburg, Bergedorfer Straße 10, 21033 Hamburg, Germany
[3]Université de Lorraine, CNRS, CRAN, 54000 Nancy, France
[4]CHRU-Nancy, Service de Neurologie, 54000 Nancy, France

BR, 0000-0002-1845-3935

Humans may be the only species able to rapidly and automatically recognize a familiar face identity in a crowd of unfamiliar faces, an important social skill. Here, by combining electroencephalography (EEG) and fast periodic visual stimulation (FPVS), we introduce an ecologically valid, objective and sensitive neural measure of this human individual face recognition function. Natural images of various unfamiliar faces are presented at a fast rate of 6 Hz, allowing one fixation per face, with variable natural images of a highly familiar face identity, a celebrity, appearing every seven images (0.86 Hz). Following a few minutes of stimulation, a high signal-to-noise ratio neural response reflecting the generalized discrimination of the familiar face identity from unfamiliar faces is observed over the occipito-temporal cortex at 0.86 Hz and harmonics. When face images are presented upside-down, the individual familiar face recognition response is negligible, being reduced by a factor of 5 over occipito-temporal regions. Differences in the magnitude of the individual face recognition response across different familiar face identities suggest that factors such as exposure, within-person variability and distinctiveness mediate this response. Our findings of a biological marker for fast and automatic recognition of individual familiar faces with ecological stimuli open an avenue for understanding this function, its development and neural basis in neurotypical individual brains along with its pathology. This should also have implications for the use of facial recognition measures in forensic science.

[†]Indicates shared first authorship.

# 1. Introduction

Neurotypical human adults can spot a familiar face—a family member, a friend, a foe or a celebrity—in a crowd of strangers at a single glance, automatically (i.e. without instruction to do so and without being able to suppress familiarity recognition). Considering that a face always appears under novel viewing conditions and often in a highly dynamic (i.e. rapidly changing) environment, this rapid individual face recognition (IFR) process constitutes an incredible achievement for the human brain, especially when the familiar face is not explicitly searched for. The apparent simplicity of IFR for familiar people masks the challenge of a task which may be performed at a high level of expertise only in the human species [1], being subtended by an extensive network of face-selective regions in the ventral occipito-temporal cortex [2–9]. Damage to this network, in particular in the right hemisphere, can lead to a rare, severe and lasting disorder of visual recognition specific to individual faces, *prosopagnosia* ([10,11]; see [12] for a recent review).

Experimental measures of IFR appeared in the mid-1960s, first as pencil-and-paper assessment tools in patients suffering IFR difficulties following right occipital and temporal brain damage [13–15], and then as behavioural tests with schematic faces or photographs to evaluate and characterize neurotypical's ability to recognize individual faces (e.g. [16,17]; see [18] for a review of early studies). A variety of computerized measures of this function were developed in subsequent decades in many laboratories, and a number of computerized behavioural tests are now widely available, together with normative data (e.g. [19–24], [25] adapted from [14]). However, despite the usefulness of these tests and decades of intense research on human face recognition, a valid and sensitive measure of human IFR is still lacking, for a number of reasons.

First, as the vast majority of experimental studies in the field of human face recognition research, behavioural tests use pictures of *unfamiliar faces* only, i.e. faces that have not yet been encoded in long-term memory in natural conditions (i.e. across many different settings and encounters). Pictures of unfamiliar faces are used for good reasons since they allow a high level of experimental control over images and a fair comparison of performance across individuals who never saw these faces before. However, while the ability to discriminate and/or match pictures of unfamiliar faces is undoubtedly an important component of the IFR function, individual face recognition is clearly not as good for unfamiliar as for familiar faces. Indeed, only the latter have been encoded in long-term memory under natural, i.e., varied and dynamic circumstances ([26]; see [27], and [28] for reviews). Artificial encoding in memory of a few images of an unfamiliar individual face, for a few seconds or minutes, as performed in a number of studies and tests, cannot compensate for this limitation. Moreover, current tests do not directly measure the recognition of a long-term familiar face identity among unfamiliar faces.

Second, typical measures of IFR rely on images that are *segmented from their natural background*, colourless and often devoid of external features (e.g. in the Cambridge face memory test, CFMT, [19]). Again, these image manipulations are done for good reasons, i.e. avoiding that discrimination/matching of pictures is accounted for by low-level visual cues or by the same specific local features. However, these manipulations also clearly reduce the ecological validity of the IFR measure.

Third, in some (sub)tests, the *exact same images* of faces have to be matched. Even when different segmented images of an individual are used (e.g. changing in size, position, head rotation or lighting direction), typical measures of IFR rely at best on two or three images of a given individual face, with a neutral expression. Such conditions do not mirror the daily life challenge of human IFR, where familiar individual faces need to be recognized across highly variable viewing conditions, sometimes against numerous (familiar and unfamiliar) distractors. Moreover, while face recognition research has mainly focused on studying visual discrimination, often in simple matching tasks, much less attention has been given to our ability to *generalize* across many visually very different images of the same person. In recent years, this issue has been rightly emphasized in human face recognition research, with studies showing a large scale of between-pictures within-person variability of the face [29,30]. A valid measure of IFR ought to take into account this large-scale within-person variability in face viewing conditions.

Fourth, in typical measures of IFR, participants are *explicitly* asked to match different pictures of an individual face against one or several distractors, to encode (unfamiliar) faces in memory, or to recognize familiar faces. This is very different from the natural conditions under which individual faces are encoded in memory, i.e. without formal training and intention to do so throughout the various encounters with these faces. Moreover, while in a number of situations we expect to meet—or we search for—specific individuals, faces are also often recognized automatically when we encounter familiar people unexpectedly (unfortunately, we cannot look at Donald Trump's face and try not to recognize it). The lack of automatic recognition of unexpected familiar individuals from their face is in

fact the primary complaint of patients with prosopagnosia. Explicit recognition tasks also carry two major disadvantages when assessing IFR. First, such tasks involve many processes, such as task understanding, motivation, attention and decisional processes, which can affect performance, drive test–retest effects and vary greatly across individuals, independently or in interaction with the specific IFR process *per se*. Second, explicit IFR tasks cannot be used, or are difficult to use as such, with all populations, for instance with infants or young children, some clinical populations, or non-human species. This prevents a fair comparison of IFR measures across these populations.

Finally, a major limitation of typical measures of IFR is the *lack of time constraints*: face stimuli are typically shown for prolonged or unlimited time in assessment tests of IFR. This is surprising because the recognition of a familiar face is usually fast in our daily experience, and it has to be fast for adequate social interactions (i.e. one can rarely scrutinize an encountered familiar face for 5 s to search for their identity). Congruent with this, behavioural and electrophysiological studies show that faces can be recognized as being familiar or having been seen before at a glance and within a few hundreds of milliseconds (e.g. [31–41]). Despite this, researchers are reluctant to add time constraints in explicit IFR tasks because these constraints can deteriorate behavioural performance even in healthy adult participants [42,43] and could even be more problematic when testing children or clinical populations. Measuring response times (RTs) in addition to task accuracy rates allows to partially circumvent this problem, but such RT measures are not included in most major behavioural tests, and are only a poor proxy of IFR processing time. Moreover, the IFR measure is then reflected in two variables, whose combination in a single measure such as (inverse) efficiency is not without limitations [44].

In summary, measures of human IFR are limited in validity at many levels. Importantly, the current limitations are not primarily due to the sole use of unfamiliar faces: studies testing behavioural recognition of natural images of highly familiar faces (i.e. celebrities or family members) (e.g. [45,46]) usually show only one (iconic) picture per face (no need for generalization across images) (but see [29]), impose only weak or no time constraints, do not require individual face discrimination (i.e. among a crowd of strangers) and require participants to explicitly recognize the famous faces.

Developing a more valid and sensitive measure of human IFR may be necessary to achieve a full theoretical understanding of this function in neurotypical human individuals along with its development, neural basis and pathology. Moreover, a more valid and sensitive measure of human IFR could be extremely valuable in forensic face recognition research, and allow a proper evaluation of other animal species or artificial systems' IFR ability through valid benchmark tests.

The present paper reports the outcome of a novel experimental approach that attempts to capture the above outlined characteristics of human IFR, i.e. a highly valid measure. The approach is based on scalp electroencephalography (EEG) and thus aims at providing a *neural* measure of IFR. Although numerous studies have compared EEG signals to familiar and unfamiliar faces (e.g. [31,32,36,37,39,41,47–52]), they presented stimuli at a slow non-periodic rate and focused on components in the time domain (event-related potentials, ERPs) to inform about the time course of this process, with various success. This is not the goal pursued here, in which electrophysiology is used to provide a valid, sensitive and objective index of recognizing a familiar face identity among variable unfamiliar faces. The approach used here takes advantage of the fairly old observation that a visual stimulus presented at a fixed rate, e.g. a light flickering on/off 17 times per second (17 Hz), generates an electrical brain wave exactly at the stimulation frequency (i.e. 17 Hz in this instance), which can be recorded over the visual cortex with electroencephalography (EEG) [53]. Such data can be transformed in the frequency domain through Fourier analysis [54], providing highly sensitive (i.e. high signal-to-noise ratio, SNR) [55] and objective (i.e. at a predetermined frequency) quantifiable markers of an automatic visual process, without explicit task. While this 'frequency-tagging' or fast periodic visual stimulation (FPVS) approach has long been confined to the study of low-level processes, i.e. ophthalmology and low-level vision (see [56] for review) as well as their modulation by spatial and selective attention (e.g. [57,58]), it has been extended in more recent years to measure visual discrimination of more complex images, in particular well-controlled images of unfamiliar faces (e.g. [59–61]).

Here we combine FPVS with EEG to directly measure the recognition of a familiar face identity among unfamiliar faces. To do so, we present highly variable *ambient* (i.e. naturally occurring) images of different unfamiliar face identities at a 6 Hz stimulation frequency, with highly variable images of the same familiar face identity inserted every seven images (i.e. 0.86 Hz). While common neural responses to all face stimuli are captured by the 6 Hz stimulation frequency, a response exactly at 0.86 Hz in the EEG spectrum would reflect a selective (i.e. differential) reliable (i.e. periodic) response to the familiar face identity (figure 1).

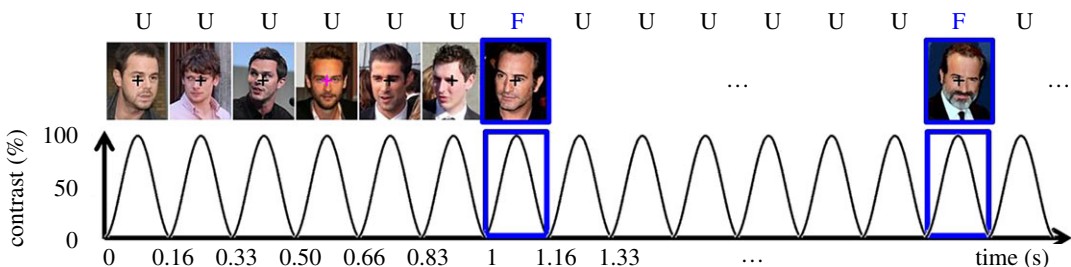

— F = 6 Hz: generic visual presentation frequency
— F/7 = 0.86 Hz: familiar individual face presentation frequency

**Figure 1.** Experimental design. Different unfamiliar face identities are repeatedly shown at 6 Hz with the same familiar face identity presented under widely different viewing conditions as every seventh image (0.86 Hz). The familiar face shown here is French actor Jean Dujardin. The other faces, also varying widely in viewing conditions, are unknown to the participants. The whole stimulation sequence lasts 74 s (including 2 s of stimulus fade-in and 2 s of fade-out). Participants have to monitor and respond to rare non-periodic colour change of the central fixation cross (200 ms, from black to red) by press the SPACE key while simultaneously monitoring the flickering stimuli. Face images shown here are with licence permits. For licence information, Blake Harrison, and Matt Johnson: Pictures licensed under the Creative Commons Attribution 2.0 Generic. Attribution: Damo 1977, and Robert Clarke, respectively. Danny Dyer, Jean Dujardin, and Tom Mison: Pictures licensed under the Creative Commons Attribution-Shared Alike 3.0 Unported. Attribution: Hilton 1949, Georges Biard, and Floatjon, respectively. Nicholas Hoult: Pictures licensed under the Creative Commons Attribution-Shared Alike 2.0 Generic. Attribution: Gage Skidmore. Jack O'Connell: Pictures licensed under the Creative Commons Attribution-Shared Alike 2.0 Generic. Attribution: gdcgraphics.

The 20 images of the same familiar individual used in a stimulation sequence show visually highly dissimilar photographs, requiring generalization. Since participants are not instructed to search for a familiar face identity, we measure implicit recognition of a familiar face identity in a stream of unfamiliar faces. We tested 16 observers in this novel paradigm, with six highly famous face identities presented in different stimulation sequences, searching for significant group indexes of the recognition of familiar face identities, but also quantifying the response to each facial identity separately. Moreover, since physical differences between images of the familiar face identity and the unfamiliar faces remain unchanged at inverted orientation but typical human observers are known to be poor at recognizing individual faces upside-down (arguably the most reliable effect in the face processing literature, shown across a wide variety of tasks, e.g. [16,62]; see [63] for review), we assessed the validity of our paradigm with upside-down images, expecting a large reduction of the IFR electrophysiological response.

## 2. Methods

### 2.1. Participants

Sixteen people (12 females, mean age = 21.31, s.d. = 1.88, range = 18–24, all right handed) took part in the experiment. They were paid 10 euros per hour as compensation upon completion of the study. All participants had normal or corrected-to-normal vision and no prior history of neurological illness.

### 2.2. Stimuli

Our stimulus set included 240 colour images (21 different face identities, see details below) of male human faces. All images showed faces of celebrities (e.g. actors, comedians, politicians) and were taken from the Internet. Only male faces were used in the study, as images of male celebrities available on the Internet appeared more variable in facial variability and expression than images of female celebrities. Images were not modified and we deliberately did not segment the faces from their original naturalistic background. We selected two sets of faces: familiar faces (i.e. well-known French celebrities that were highly familiar to our participants due to exposure through television, press or social media) and unfamiliar faces (i.e. less well-known British celebrities that were unfamiliar to our participants as assessed by familiarity questionnaires) (see examples in figure 1).

Six different 'famous' French face identities served as *the familiar faces appearing periodically* (see procedure below), one identity per stimulation sequence. The six familiar faces included in this study were the ones with

the highest recognition rates taken from a larger set of celebrities, here the actors *Gerard Depardieu*, *Dany Boon*, *Jean Dujardin*, *Gad Elmaleh* and *Vincent Cassel*, and the politician *Nicolas Sarkozy*. We then randomly chose 15 British face identities who served as our *unfamiliar faces* (see figure 1 for examples). We selected a varied set of images for each of our identities: 20 different images for each of our six familiar face identities (120 images) and eight different images for each of our 15 unfamiliar face identities (120 images). Face identities were shown across naturally occurring images (i.e. changes in lighting, view, appearance and age) with the only restriction that faces were clearly recognizable and their head filled the entire image. Images were all 255 pixels in height, while image width varied ($190 \pm 30$ pixels) to preserve variations in head width of different identities and orientations. Given that head width was slightly but significantly larger on the natural images of *Depardieu* and *Boon* than for unfamiliar images ($ps < 0.01$; no significant difference for any of the other four identities with unfamiliar faces, $ps > 0.1$), slightly wider images were used for these two personalities in order to minimize differences in head versus background surface areas across facial identities. This way, head area/image area ratios did not differ across the six familiar faces and for each face identity with respect to the unfamiliar faces ($ps > 0.1$). Control for such low-level stimulus variations was fully afforded by showing the exact same image set upside-down ($180°$ rotation) in the experiment. Every image was shown an equal number of times throughout every 70 s stimulation sequence (i.e. three times). This resulted in a total of 480 images shown in the experiment (half of them shown in the upright and the other half in the inverted orientation).

# 3. Procedure

## 3.1. Questionnaires, pre- and post-EEG testing

To assess subjects' familiarity with our image set, participants filled in two paper-and-pencil familiarity questionnaires, one prior to and one following EEG testing. Prior to testing, the *Names Questionnaire* (Q1) showed a list of all 21 celebrities included in the study with their respective profession. Participants were asked to indicate whether or not they knew a person (Yes/No) and if they answered positively, how well they could mentally visualize this person's face (5-point Likert scale from 1 = poorly to 5 = very well). After testing, the *Images Questionnaire* (Q2) was administered showing a list of 21 images (one exemplar image was randomly taken from the images shown in the experiment) corresponding to the names shown in Q1 to which participants responded with Yes/No answers to '*I know this person*'. If participants knew a person's face, they were asked to write down this person's name and profession and to give a rated response (1–5 point Likert scale) to '*I have seen this person's face before (tv etc)*' with 1 = few times and 5 = many times. Participants were also asked to report whether they had seen this person during the experiment (Seen/Not Seen). In both familiarity questionnaires, names and images were presented in a random order.

## 3.2. Visual stimulation

In the main experiment, participants sat at a table in a dimly lit room in front of a computer monitor at a viewing distance of 80 cm (screen resolution of 1280 pixels $\times$ 1024 pixels at a frame rate of 120 Hz). During stimulation, all stimuli appeared in the centre of a uniform light grey background subtending on average $9.09° \times 6.8°$ of visual angle. Each stimulation sequence consisted of a 2–5 s pre-stimulation period in which the participant viewed only the central fixation cross, and a 74 s stimulation sequence during which images were presented at a fast periodic rate of 6 Hz (six images per second) (figure 1). Images were presented with a customized stimulation software programmed in Java that sinusoidally modulated the contrast of each image from 0% to 100% to 0% (e.g. [59,60]; figure 1). In addition, the stimulation gradually increased from 0% to 100% contrast over the first 2 s of stimulation (fade-in) and decreased accordingly over the last 2 s of stimulation (fade-out), keeping the sinusoidal contrast modulation. These fading periods were intended to avoid abrupt EEG responses at the beginning and end of stimulation (onset and offset visual evoked potentials, surprise reactions from participants or blinks). Participants were instructed to fix their eye gaze on a central fixation cross ($0.31° \times 0.31°$ visual angle) that appeared in the middle of the images throughout the stimulation sequence, and to press the SPACE key whenever the fixation cross changed colour (200 ms, black to red; 15 change occurrences in each stimulation sequence, at random times), while continuously monitoring the flickering stimuli.

There were 12 different stimulation sequences (six with upright images and six with inverted images, random order) of 70 s of stimulation each and each sequence was repeated once (24 sequences in total).

Unfamiliar faces (different identities and different images) were presented at the predefined frequency of approximately 6 Hz, which ensured only one gaze fixation on each face, and was also selected based on previous studies showing the largest occipito-temporal response to trains of different unfamiliar faces at around 6 Hz [64] as well as on pilot data of the present experiment. In each sequence, different images of the same familiar face identity were presented as every seventh images (replacing an unfamiliar face) providing a second predefined frequency of 6 Hz/7 ~0.86 Hz (0.8571 Hz). At 0.86 Hz, the temporal distance between two images of the familiar face identity was 1167 ms, ensuring that specific responses to familiar faces would not overlap with one another. Familiar faces always showed the same familiar face identity within each individual 70 s sequence.

In line with previous studies, presenting face images at 6 Hz should elicit a robust EEG response at 6 Hz (e.g. [64]) but also at harmonics (12 Hz, 18 Hz, etc.) of this base frequency rate. Harmonics are due to stimulation constraints (i.e. the input is not perfectly sinusoidal due to image complexity and refresh rate limitations, with frames of 8.33 ms at 120 Hz) and most importantly the nonlinearity of the neural response [55,56,61]. This 6 Hz and harmonics response reflects general visual-stimulation responses to faces, irrespective of their familiarity. In addition, and critically, if the brain automatically discriminates the familiar face identity from the unfamiliar faces and if this discrimination generalizes across large changes in within-person image variability, there should be a significant individual face recognition response at 0.86 Hz and harmonics (1.71 Hz, 2.57 Hz, etc.).

## 3.3. EEG recording

The electroencephalogram (EEG) was acquired using a 128-channel Biosemi Active 2 system (BioSemi, Amsterdam, The Netherlands), with electrodes including standard 10–20 system locations as well as additional intermediate positions (http://www.biosemi.com/headcap.htm, relabelled to more conventional labels of the 10–5 system, see supplementary fig. S1 in [65]). The EEG was sampled at 512 Hz. Electrode offset was reduced to under ± 20 mV for each individual electrode by softly abrading the scalp underneath with a blunt plastic needle and injecting the electrode with saline gel. Eye movements were monitored by four additional electrodes placed at the outer canthi of the two eyes, and above and below the right orbit. During the experiment, triggers were sent via parallel port from the stimulation computer to the EEG recording computer at the beginning and the end of each stimulation sequence, and at the minima (0% contrast) of all 6 Hz stimulation cycles (i.e. onsets/offsets of images). The temporal synchrony between the trigger and the stimulus onset was verified by a photodiode prior to the experiment. Recordings were manually initiated by the experimenter when participants showed artefact-free EEG signals.

# 4. EEG analysis

## 4.1. Preprocessing

All EEG data were analysed using the free software *Letswave 5* (https://github.com/NOCIONS/Letswave5) running on Matlab. Data analysis was similar to many previous studies using similar FPVS paradigms with face and object images or segmented images of unfamiliar faces (see [61]), but is nevertheless described in full here. For each individual participant dataset, the raw EEG response was band-pass filtered between 0.1 Hz and 100 Hz (fourth-order low-pass Butterworth filter). EEG data were then segmented to include 2 s before and 2 s after each 74 s sequence, resulting in 78 s segments (−2 s to 76 s). Data files were then downsampled to 256 Hz to reduce file size and data processing time. Then, noisy and artefact-ridden channels (fewer than 5% of 128 channels, i.e. a maximum of six channels) containing deflections larger than 100 μV in multiple presentation sequences were rebuilt using linear interpolations from immediately adjacent noise-free channels. All channels were re-referenced to the common average of the 128 scalp channels. EEG recordings were then segmented again from stimulation onset (after 2 s fade-in) until 70.83 s, corresponding exactly to 63 complete cycles at 6 Hz (17 622 bins), in order to avoid spectral leakage.

## 4.2. Frequency domain analysis

In the main analysis, for each participant, we collapsed individual stimulation sequences across the six famous face identities in the time domain to reduce EEG activity that is not phase-locked to the stimulus. Sequences with upright and inverted faces were averaged separately. A FFT was applied to

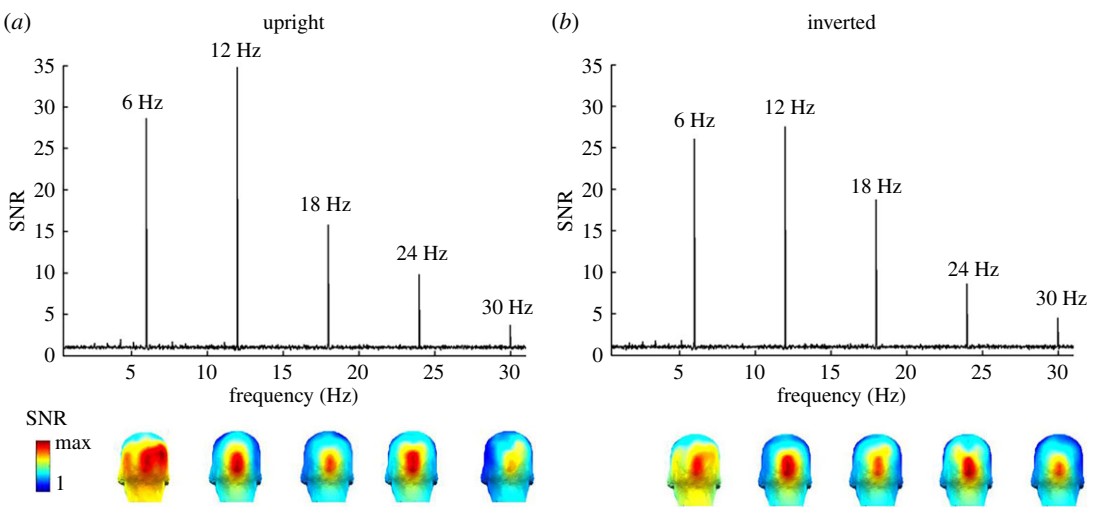

**Figure 2.** Grand-averaged EEG spectra in SNR over the medial occipital region of interest (ROI) (averaged across channels POOz, Oz, OIz, and Iz). A SNR of 35 corresponds to 3400% of amplitude increase relative to EEG noise level. Responses at 6 Hz and subsequent harmonics (e.g. 12 Hz, 18 Hz, 24 Hz and so forth), shown separately for (*a*) upright and (*b*) inverted face stimuli. Three-dimensional scalp maps are shown below for significant harmonics, showing consistent topographies across face orientation conditions. Colour scale indicates the range from 1 to the maximum SNR for each individual harmonic.

the sequence-averaged data segments, and an amplitude spectrum (normalized by $N/2$, in µV) was extracted in the frequency domain (ranging from 0 to 256 Hz) for each channel as a normalized amplitude spectrum (µV) in the frequency domain. Each spectrum had a high frequency resolution (i.e. distance between two adjacent frequency components) of 0.0141 Hz, which is the inverse of the segment duration (70.83 s). This allowed unambiguous identification of the frequencies of interest (0.86 Hz and harmonics). In a complementary analysis, the FFT was also applied separately for each face identity sequence (averaged data segment across two sequences with the same familiar face identity presented).

EEG noise was estimated as in previous studies as the average amplitude of the 20 bins surrounding the frequency bins of interest (10 bins on each side, [61,66]). The noise calculation excluded the immediately adjacent bins in case of resultant spectral leakage, and the local maximum and minimum amplitude bins to avoid projecting the signal into the noise during EEG spectrum baseline correction. Two methods of baseline correction were applied: *divide baseline* to display the data as signal-to-noise ratio (SNR) spectra, and *subtract baseline* (SB) so that the baseline-corrected response at each frequency bin was then expressed in microvolts (µV). The baseline-division (SNR) method is used for better visualization of the response (especially small responses at high-frequency harmonics, or the first harmonic response, where EEG noise is large; e.g. [67]), while the baseline-subtraction method is used for quantification of the response of interest by summing the harmonics and for statistical analysis [61]. To illustrate the response differences between two face orientation conditions, we grand-averaged the amplitude spectra (in µV) in the frequency domain across all channels and all participants for each orientation condition, and computed SNR spectra (figures 2 and 3).

In order to identify significant responses at the relevant stimulation frequencies (i.e. 6 Hz, 0.86 Hz, and their harmonics), we averaged the amplitude spectra across all subjects and electrodes across the two orientation conditions and computed a $z$-score at each discrete frequency bin. Our baseline for $z$-score calculation was also the mean amplitude of 20 frequency bins neighbouring the frequency of interest, excluding the immediately adjacent bins and the local maximum and minimum amplitude bins (e.g. [59,61,66]). To take into account the multiple harmonics tested, we considered responses with a $z$-score greater than a conservative threshold of 3.1 to be significant ($p < 0.001$, one tailed, i.e. signal greater than noise). The response was then quantified separately for each condition and electrode by summing across all significant harmonic frequencies in baseline-corrected amplitude as a quantitative measure of the size of the response at our two stimulation frequencies [61].

For the analysis at the individual level, a significant individual face recognition response was identified by summing the raw amplitudes of significant frequency harmonics (as determined on grand-averaged data), and computing $z$-scores based on mean amplitude of neighbouring bins (again 20 on each side, excluding the immediately adjacent bins and local maximum and minimum amplitude bins; [60,61,64]). Threshold of significance was placed at a $z$-score of 3.1 ($p < 0.001$, one tailed).

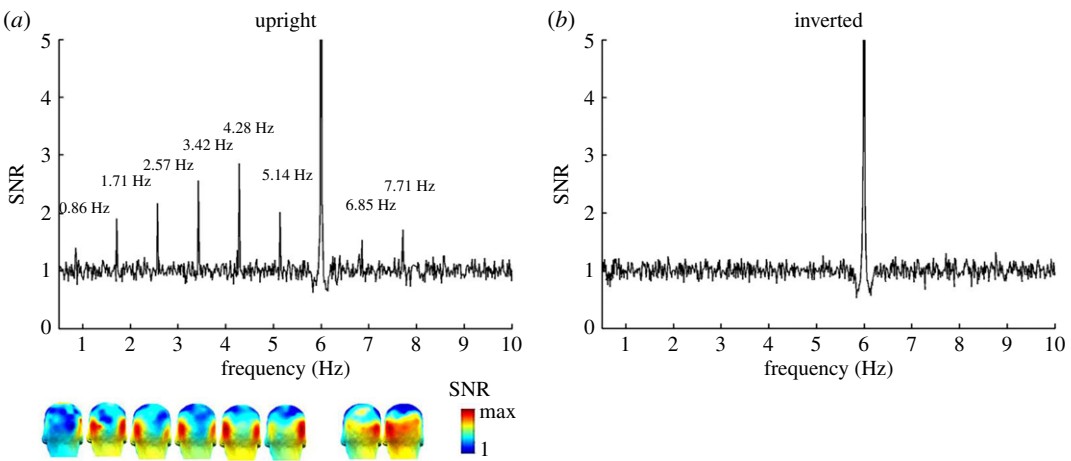

**Figure 3.** Grand-averaged EEG spectra in SNR over the occipito-temporal ROI (averaged across left and right occipito-temporal channels, P9 & 10, PO11 & 12 respectively). A SNR of 3 corresponds to 200% of amplitude increase relative to EEG noise level. Responses at 0.86 Hz and subsequent harmonics (e.g. 1.71 Hz, 2.57 Hz, 3.42 Hz and so forth) shown separately for (*a*) upright and (*b*) inverted face stimuli. Three-dimensional scalp maps are shown for significant harmonics, indicating the largest responses over left and right occipito-temporal channels. On grand-averaged data, there were no significant individual harmonics for inverted faces in this paradigm. Note that the relatively small SNR at 0.86 Hz for upright faces relative to higher harmonics does not imply that amplitude at that frequency is low, but to a larger EEG noise in this low frequency range (see also e.g. [60]). The colour scale is showing the scale from 1 to the maximum SNR of each harmonic. Note that the scale of *y*-axis is cut to 5 in order to have a better visualization of the SNR of each significant harmonic of the individual face recognition responses. The response SNR at the presentation rate of 6 Hz is much higher than 5 (figure 2).

We statistically tested neural discrimination responses at two levels: across the whole scalp (average across all electrodes) and at local regions of interest (ROIs) where both the amplitudes at 6 Hz or 0.86 Hz reached their maxima, as in previous FPVS studies measuring categorization of faces versus objects or unfamiliar individual face discrimination (e.g. [60,61,65,68]). ROIs were defined by averaging the four electrodes with the largest summed-harmonic response across conditions. Therefore, we focused on a medial occipital ROI (average of central midline electrodes OIz, Iz, Oz and POOz) where the general visual response to faces (irrespective of orientation) was maximal, and a bilateral occipito-temporal ROI (OT ROI; average of posterior electrodes P9 and PO11 for left hemisphere and average of P10 and PO12 for right hemisphere) where individual recognition responses to familiar faces were maximal.

# 5. Results

## 5.1. Behavioural results

### 5.1.1. Fixation cross task

Response times (RTs) were calculated relative to the onset of target fixation cross. Analyses were based on the mean of correct RTs. Responses were considered correct if they occurred between 150 ms and 1000 ms following target onset. The results showed that accuracy rates for the fixation cross colour change task was close to ceiling in the two conditions (upright: $M = 98.55\%$, s.d. $= 1.86\%$; inverted: $M = 99.35\%$, s.d. $= 0.77\%$; $t_{15} = -1.79$, $p = 0.094$). There was no difference in correct RTs between the two conditions (upright: $M = 398$ ms, s.d. $= 41$ ms; inverted: $M = 396$ ms, s.d. $= 36$ ms; $t_{15} < 1$).

### 5.1.2. Familiarity questionnaires

#### 5.1.2.1. Name questionnaire (Q1, before EEG testing)

On average, participants showed perfect recognition rates (100%) for the names of five French celebrities, *Gerard Depardieu, Dany Boon, Jean Dujardin, Gad Elmaleh* and *Nicolas Sarkozy* and 63% recognition accuracy for less well-known French actor *Vincent Cassel*. Recognition of British celebrities' names (considered as 'unfamiliar' to our tested participants) was low (6.2%, on average less than 1 of the 15 identities per participant). Visualization scores for the familiar faces were high: 4.5 on average (with 1 = poorly,

5 = well). Visualization was the highest for Jean Dujardin = 4.7 followed by Gerard Depardieu = 4.6, Nicolas Sarkozy = 4.6, Dany Boon = 4.6 and Gad Elmaleh = 4.5, while Vincent Cassel had the lowest visualization score, 4.1. In comparison, the average visualization score for the few known 'unfamiliar' British celebrities was 2.4.

### 5.1.2.2. Image questionnaire (Q2, after EEG testing)

Overall, recognition of the exemplar familiar face image was at ceiling (100%) for five of our six celebrities, *Gerard Depardieu, Dany Boon, Jean Dujardin, Gad Elmaleh* and *Nicolas Sarkozy*. Recognition rates dropped to 81% for *Vincent Cassel*. Participants recognized 8.3% of all exemplar 'unfamiliar' faces (on average less than 2 of the 15 identities per participant). Participants indicated having seen faces of known celebrities numerous times (4.2 on average with 1 = few, 5 = many). Nicolas Sarkozy was the 'most seen' celebrity with a score of 4.6 followed by Jean Dujardin = 4.5, Gerard Depardieu = 4.4, Dany Boon = 4.4, Gad Elmaleh = 4.2 and Vincent Cassel = 3.5. In comparison, the average exposure score for the few known 'unfamiliar' faces was 2.0.

Only two participants indicated to recognize both name and image of a supposedly 'unfamiliar' face. Interestingly, of the familiar faces, only *Gerard Depardieu* and *Dany Boon* were reportedly 'seen' by all participants during the EEG experiment. When asked to name the most famous out of the six celebrities, most participants chose *Nicolas Sarkozy* (73%), while *Gerard Depardieu* was named as the easiest to recognize (66%).

## 5.2. EEG results

### 5.2.1. Frequency-domain indexes of generic visual responses

The base stimulation rate of 6 Hz generated a significant ($p < 0.001$) response at the first five consecutive harmonics (i.e. 6 Hz, 12 Hz, and so forth). Responses at these five harmonics were significant on both upright and inverted images (both $ps < 0.001$, see figure 2). Interestingly, for upright faces only, the first harmonic at 6 Hz was associated with a right occipito-temporal response in addition to the medial occipital focus. All harmonics for inverted faces, and the remaining harmonics (from 12 Hz on) for both orientations were associated with a medial occipital topography (figure 2).

Although the reduction of signal for inverted faces compared to upright faces was relatively small (i.e. 14%), the sum of baseline-subtracted amplitude values across these five frequencies was still significantly larger for upright versus inverted images (average all channels; upright: $M = 1.40\,\mu V$, s.d. $= 0.40\,\mu V$; inverted: $M = 1.20\,\mu V$, s.d. $= 0.40\,\mu V$; $t_{15} = 4.12$, $p < 0.01$). Responses at the first five consecutive harmonics over the medial occipital ROI were maximal, reflected again by a larger response to upright compared to inverted sequences (upright: $M = 4.25\,\mu V$, s.d. $= 2.17\,\mu V$; inverted: $M = 3.70\,\mu V$, s.d. $= 1.98\,\mu V$; $t_{15} = 3.57$, $p < 0.01$).

Given the difference of scalp topography, we also tested the response at the first harmonic independently of the remaining harmonics. At 6 Hz, difference between the two face orientation conditions was compared over different ROIs, i.e. the OT ROI (P9 & 10, PO11 & PO12) and the medial occipital ROI (OIz, Iz, Oz, and POOz). At the OT ROI, the 6 Hz response was significantly larger for upright as compared to inverted faces (upright: $M = 1.77\,\mu V$, s.d. $= 0.71\,\mu V$; inverted: $M = 1.46\,\mu V$, s.d. $= 0.7\,\mu V$; $t_{15} = 2.25$, $p < 0.05$). However, at the medial occipital ROI, there was no difference between the two face conditions (upright: $M = 1.82\,\mu V$, s.d. $= 1.32\,\mu V$; inverted: $M = 1.69\,\mu V$, s.d. $= 1.29\,\mu V$; $t_{15} = 1.17$, $p > 0.1$). For the sum of remaining significant harmonics (harmonics 2–5), measured over the medial occipital ROI, the response to upright faces was still significantly larger than to the inverted faces (upright: $M = 2.44\,\mu V$, s.d. $= 1.26\,\mu V$; inverted: $M = 2.02\,\mu V$, s.d. $= 1.18\,\mu V$; $t_{15} = 5.12$, $p < 0.001$).

### 5.2.2. Individual face recognition response

On grand-averaged data across all channels and conditions, we observed significant responses at the familiar identity stimulation frequency of 0.86 Hz and its harmonics (e.g. 1.71 Hz, 2.57 Hz, 3.42 Hz, and so forth), with up to eight significant harmonics (until 7.71 Hz, skipping the 6 Hz base rate). This response was driven essentially by upright faces, with the highest SNR values being found, as expected, over occipito-temporal channels (figure 3a). Strikingly, responses were virtually absent when the same face images were rotated by 180°, with no harmonics reaching significance level, and SNR over occipito-temporal channels being close to 1 (figure 3b). These observations show that the familiar

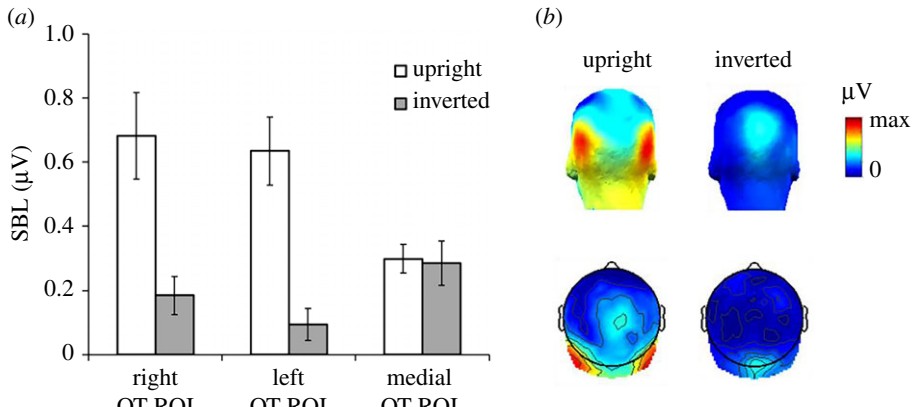

**Figure 4.** (a) Sum of 0.86 Hz response and significant harmonics (in μV) plotted separately for upright and inverted sequences at the two OT ROIs, right versus left hemisphere, and the medial occipital ROI. (b) Three-dimensional (top) and two-dimensional (bottom) topographic scalp maps of the two face orientation conditions for the individual face recognition response. The colour scale indicates the range from 0 to the maximum amplitude (in μV) across two face conditions.

face identity was consistently (i.e. periodically) discriminated from the unfamiliar faces at upright orientation, but not, or only weakly, at inverted orientation.

Comparing the sum of baseline-subtracted amplitudes of harmonics 1–6, 8 and 9, EEG amplitude across the whole scalp was significantly larger for upright images ($M = 0.20\,\mu V$, s.d. = $0.15\,\mu V$) as compared to inverted images ($M = 0.06\,\mu V$, s.d. = $0.1\,\mu V$, $t_{15} = 3.25$, $p < 0.01$). At all posterior ROIs, the response to upright faces was significantly above zero (right: $M = 0.68\,\mu V$, s.d. = $0.52\,\mu V$, $t_{15} = 5.23$, $p < 0.001$; left: $M = 0.64\,\mu V$, s.d. = $0.41\,\mu V$, $t_{15} = 6.21$, $p < 0.001$; medial occipital: $M = 0.3\,\mu V$, s.d. = $0.17\,\mu V$, $t_{15} = 6.87$, $p < 0.001$), and there were weak but significant responses also for inverted faces (right: $M = 0.18\,\mu V$, s.d. = $0.24\,\mu V$, $t_{15} = 3.1\,\mu V$, $p < 0.01$; left: $M = 0.09\,\mu V$, s.d. = $0.18\,\mu V$, $t_{15} = 2.14$, $p < 0.05$; medial occipital: $M = 0.29\,\mu V$, s.d. = $0.27\,\mu V$, $t_{15} = 4.3$, $p < 0.001$). We then conducted a within-subject ANOVA on the sum of baseline-subtracted amplitudes at bilateral occipito-temporal ROIs with the factors *Orientation* (upright versus inverted) and *Hemisphere* (right versus left) (figure 4a). There was a highly significant main effect of *Orientation* (upright: $M = 0.65\,\mu V$, s.d. = $0.10\,\mu V$; inverted: $M = 0.13\,\mu V$, s.d. = $0.04\,\mu V$; $F_{1,15} = 20.60$, $p < 0.001$), reflecting a large response reduction for the inverted faces, by a factor of 5. There was no effect of *Hemisphere* and no significant interaction (both $Fs < 1.7$). However, analyses at the medial occipital ROI showed no effect of *Orientation* (upright: $M = 0.29\,\mu V$, s.d. = $0.17\,\mu V$; inverted: $M = 0.28\,\mu V$, s.d. = $0.26\,\mu V$; $t_{15} = 0.164$, $p = 0.87$).

### 5.2.3. Specific familiar face identities

We then considered the IFR responses separately for each of the six familiar face identities (one shown per sequence, one repetition per identity). On grand-averaged data, figure 5 shows clear responses for upright sequences over OT ROI for all six familiar faces, i.e. *Gerard Depardieu, Dany Boon, Jean Dujardin, Gad Elmaleh, Nicolas Sarkozy* and *Vincent Cassel*. EEG amplitudes varied as a function of identity, however, with the largest response to *Depardieu* ($M = 1.13\,\mu V$, s.d. = $0.84\,\mu V$). IFR responses were still high for *Boon* ($M = 0.70\,\mu V$, s.d. = $0.58\,\mu V$), *Dujardin* ($M = 0.64\,\mu V$, s.d. = $0.58\,\mu V$) and *Sarkozy* ($M = 0.58\,\mu V$, s.d. = $0.50\,\mu V$), and gradually decreased for *Elmaleh* ($M = 0.39\,\mu V$, s.d. = $0.50\,\mu V$) and *Cassel* ($M = 0.30\,\mu V$, s.d. = $0.40\,\mu V$). In contrast, IFR responses to inverted familiar faces were considerably smaller (figure 5b), highlighting the detrimental effect of inversion on familiar face recognition. The largest response was elicited to *Depardieu* ($M = 0.49\,\mu V$, s.d. = $0.36\,\mu V$), followed by *Boon* ($M = 0.21\,\mu V$, s.d. = $0.35\,\mu V$), *Sarkozy* ($M = 0.19\,\mu V$, s.d. = $0.41\,\mu V$) and *Elmaleh* ($M = 0.17\,\mu V$, s.d. = $0.30\,\mu V$). Responses were smallest for *Dujardin* ($M = 0.04\,\mu V$, s.d. = $0.35\,\mu V$) and absent for *Cassel* ($M = -0.03\,\mu V$, s.d. = $0.35\,\mu V$).

Statistical analyses confirmed significant IFR responses (z-scores greater than 3.1, $p < 0.001$, z-score range = 4.1–14.1) for each of the six familiar identities on upright sequences, averaged across participants at OT ROI. We also observed significant, but considerably smaller posterior responses to three out of six familiar identities on stimulation sequences with inverted faces (i.e. *Depardieu, Elmaleh* and *Sarkozy*, z-score range = 1.9–8).

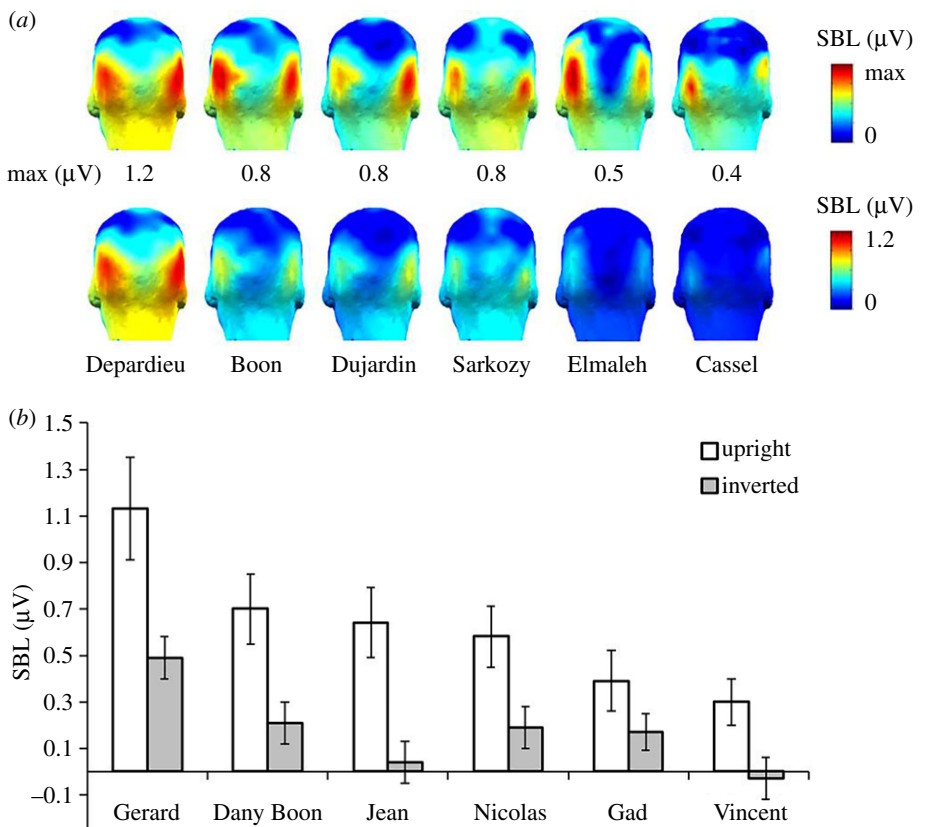

**Figure 5.** Sum of baseline-corrected responses at 0.86 Hz and harmonics (in μV) shown individually for each of the six familiar face identities over OT ROI. (*a*) Scalp maps for each face identity (individual maxima upper panel, same scale lower panel) on upright sequences and (*b*) mean amplitudes to individual identities for the two types of trials.

A repeated-measures ANOVA with factors *Face Identity* (six levels) and *Orientation* (upright versus inverted) at OT ROI showed a main effect of *Face Identity* ($F_{5,75} = 9.03$, $p < 0.001$) confirming the clear difference in the size of the neural response across the six face identities. There was also a main effect of *Orientation* ($F_{1,15} = 21$, $p < 0.001$), showing a larger response amplitude for the upright faces ($M = 0.66$ μV, s.d. $= 0.37$ μV) than the inverted faces ($M = 0.18$ μV, s.d. $= 0.14$ μV). The interaction of *Orientation × Face Identity* was not significant ($F_{5,75} = 1.1$, $p = 0.36$). Further pairwise analysis showed that responses to Gerard Depardieu was significantly larger than almost all other face identities (Jean Dujardin: MD $= 0.47$, $p < 0.01$; Nicolas Sarkozy: MD $= 0.43$, $p < 0.05$; Gad Elmaleh: MD $= 0.53$, $p < 0.05$; Vincent Cassel: MD $= 0.67$, $p < 0.01$), except Dany Boon (MD $= 0.35$, $p > 0.1$). Responses to Nicolas Sarkozy was also larger than to Vincent Cassel (MD $= 0.21$, $p < 0.05$). All multiple comparisons were adjusted with Bonferroni correction.

### 5.2.4. Individual participants

We also analysed EEG data at the individual participant level. There were clear individual face recognition responses to familiar faces in the majority of participants at the OT ROI. Figure 6 shows summed baseline-subtracted amplitude spectra across 0.86 Hz and harmonics for upright faces, separately for each of the 16 subjects. We quantified these responses by calculating *z*-scores individually for each participant over OT ROI for upright and inverted sequences, respectively.

For *upright* images, 14 out of 16 individuals showed clear familiar IFR responses over OT ROI. Thirteen subjects showed a highly significant effect over bilateral OT regions (*z*-score $> 3.1$, $p < 0.001$, range $= 3.3–18.4$), and one subject had a *z*-score of 1.68 ($p < 0.05$). Two participants showed no effect (*z*-score $< 1$). With *inverted* images, only 6 out of 16 subjects showed a significant response over OT electrodes (*z*-score greater than 1.64, $p < 0.05$, range $= 2–4.9$). The remaining ten subjects did not show a significant response.

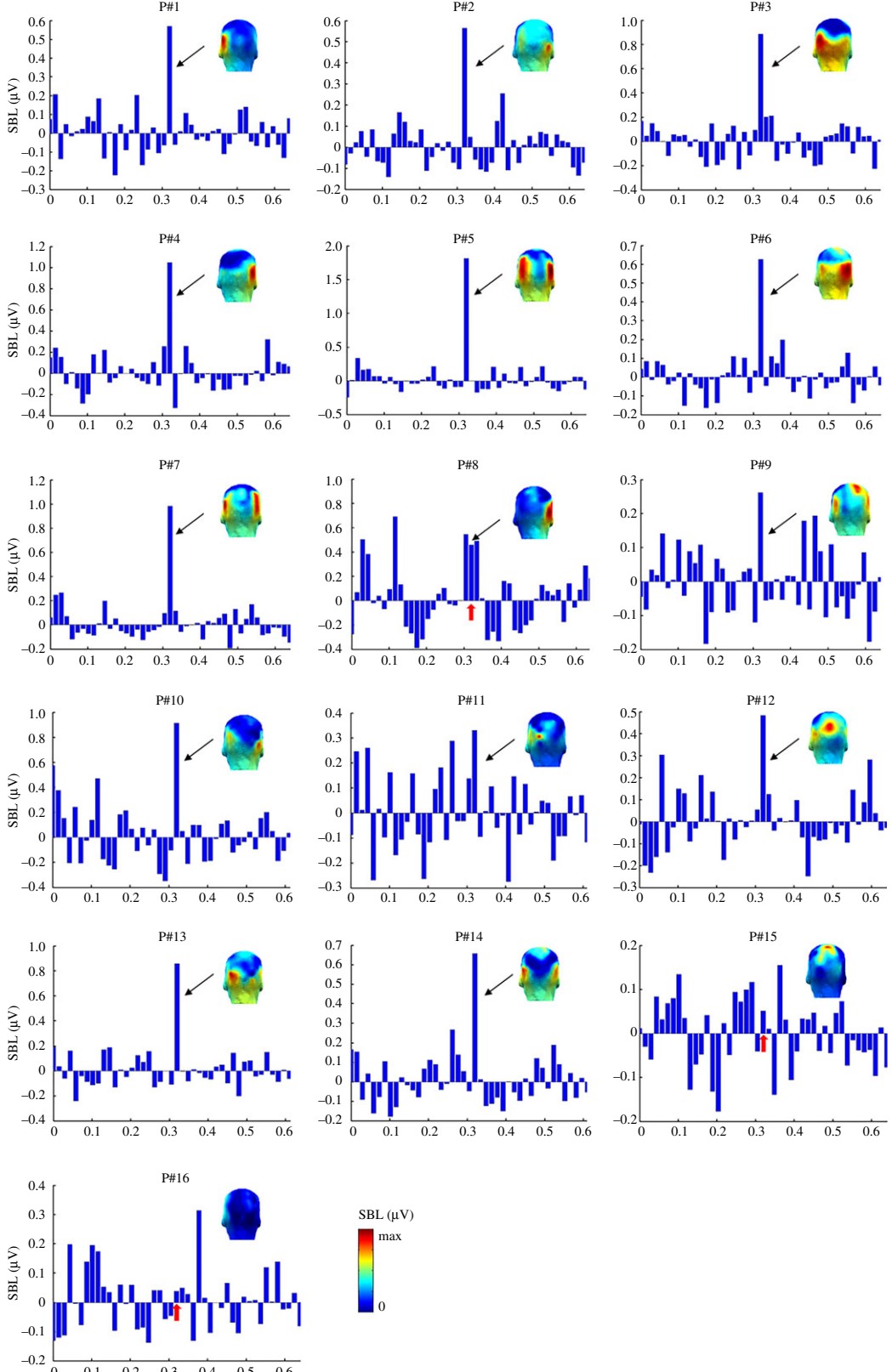

**Figure 6.** Topographic scalp maps and bar graphs of summed amplitudes (in μV) of baseline-corrected mean amplitudes (sum at 0.86 Hz and harmonics) at the OT ROI (averaged across 4 channels: P9 & 10, PO11 & 12) for upright images, shown separately for individual participants. Each subplot is showing the amplitudes of 45 bins with the middle one indicating the frequency bin of our interest (i.e. 0.86 Hz). The unit of the x-axis of each subplot is arbitrary. Note that P15 and P16 did not show a significant response at the OT ROI. The colour scale indicates the range from 0 to the maximum baseline-corrected amplitude for each subject.

# 6. Discussion

Combining FPVS and EEG, we tested whether neural responses of individual face recognition (IFR) are elicited to familiar faces shown periodically at 0.86 Hz among unfamiliar faces presented at a generic 6 Hz stimulation frequency. To resemble natural conditions, the same familiar individual appeared across 20 highly variable images, so that a 0.86 Hz response reflects the brain's ability to generalize across various images of the same familiar identity.

We observed neural responses at the general 6 Hz stimulation frequency, a mixture of low- and high-level processes, with the largest response observed over the right occipito-temporal cortex at the first harmonic (6 Hz) for upright but not inverted faces. Higher harmonic responses (12 Hz, 18 Hz, etc.) for both orientations focused on the medial occipital region (figure 2). This pattern of response resembles observations made in previous FPVS-EEG studies when trains of different unfamiliar faces are presented at frequency rates ranging from 3.5 to 6 Hz [59,64,66,69]. Yet, the use of widely variable natural images of unfamiliar faces here for the first time appears to increase the contrast between the scalp topographies associated with first harmonic response at 6 Hz and the higher harmonics (figure 2). The medial occipital topography of higher harmonics suggests that they essentially reflect low-level visual processes, as typically observed with frequency rates of face stimuli above 8–10 Hz [64]. Most importantly for the present study, within a few minutes of recording, we observed clear EEG responses at 0.86 Hz to the periodically appearing same familiar face identity over bilateral occipito-temporal regions. We take this latter response as an index of the fast and automatic recognition of a familiar individual face among a crowd of unfamiliar faces.

Over the past three decades, many studies have compared electrophysiological responses (ERPs) recorded on the human scalp to pictures of familiar and unfamiliar faces (e.g. [31,32,36,37,39,41,47–52]). Most recently, some studies have presented unfamiliar and familiar(ized) faces in different streams at rapid periodic rates, measuring 'steady-state visual evoked potentials' (SSVEPs) to the two types of stimuli [70,71]. Compared to these studies, the present approach and its findings differ on a number of aspects. First, the 0.86 Hz response in the EEG spectrum does not reflect the mere response to the sudden appearance of the familiar face identity, but the *differential* neural response between that familiar identity and the unfamiliar faces. Indeed, in this kind of 'oddball-like' paradigm, the common neural response to the familiar and unfamiliar faces is reflected in the 6 Hz response and its harmonics (12 Hz, etc.), while a 0.86 Hz response emerges only if a population of neurons respond *specifically* (or differently) to the repeated familiar face identity (e.g. [65] and [61] for a similar logic in measuring selective responses to natural images of faces as a category; [72] for differential responses between words and pseudowords). The present paradigm therefore captures a *direct* familiar individual face recognition response (i.e. no *post hoc* subtraction is required between separate responses to unfamiliar and familiar faces). Second, due to the high frequency resolution provided by analysing long temporal windows, the IFR response is expressed in the EEG in very narrow frequency bins, providing high objectivity (i.e. appearing exactly at frequency rates predicted by the experimenter) and high sensitivity (i.e. high SNR) despite testing participants only for a few minutes. Third, we argue that the paradigm has high validity because it is based on a measure obtained with natural images (very rarely used in the EEG studies above, e.g. [37,41]), at a rate allowing only one fixation per face (i.e. single-glance recognition). While this time constraint (i.e. each familiar face identity stimulus is forward- and backward-masked by unfamiliar face images) may appear too severe compared to real-world conditions, it is not unlike—and could even be slower—than the rate at which familiar faces could appear while exploring a static or dynamic crowd in a visual scene. Moreover, behavioural studies indicate that familiar(ized) faces can be recognized at a high level of accuracy only from a single glance [33,38,40], even when appearing in a train of unfamiliar faces [73]. Importantly, this time constraint can be applied here without an increase of decisional errors because there is no requirement of explicit recognition from the participants, i.e. the paradigm measures automatic face recognition. Also, and important for ecological validity, the wide physical variability among the distractors (i.e. unfamiliar faces) and the various images of each specific famous identity ensure that the measures include two simultaneous processes of IFR: *discrimination* ('telling faces apart') and *generalization* ('telling faces together'). Variability in photos of the same face is a topic that has, until recently, been somewhat overlooked in face recognition research [30,74], in part because it is challenging to include numerous widely variable images of the same face identity in paradigms measuring behavioural or neural activity. Yet, studying IFR across ambient, natural and variable images is important to capture the intrinsic challenges of individual face recognition in natural

settings. One of the main tasks of our visual system is to 'see through' visually different image properties and to learn to detect those features of an individual's face that are stable and invariant to any dynamic changes (e.g. expression or view; see [75]). This is in line with recent research suggesting that learning new faces or their structural properties is facilitated by exposure to very different images of the same individual, i.e. high image variability [76]. Here we provide a neural measure that does not only take into account this image variability but takes advantage of periodicity to force generalization over images varying widely in low-level physical features.

The validity of the present paradigm is reinforced by the large effect of stimulus inversion. When face images were rotated by 180°, neural IFR responses at 0.86 Hz were strongly reduced, i.e. by a fivefold factor. Moreover, this reduction was specific to occipito-temporal electrodes, with no reduction on medial occipital channels. We observed this inversion effect (i.e. larger responses to upright than inverted stimuli) across each of our six familiar face identities (figure 5). This decrease is in line with the drop of performance observed in a variety of behavioural tasks measuring different aspects of IFR (often with unfamiliar faces, e.g. [63] for review; e.g. [40,77] for behavioural effects on familiar face recognition). It also aligns with decreases of neural individual face discrimination indexes for pictures of unfamiliar faces in humans [59,60,78–81]. Yet, we note that despite using natural images with all external cues preserved, the inversion effect is particularly large in the present paradigm, an effect that we attribute to two main factors. First, as in a similar 'oddball-like' FPVS-EEG paradigm showing large inversion effects for unfamiliar face discrimination [60,82], the tight time constraints imposed in the paradigm, i.e. one fixation per face, prevent a detailed analysis of local facial features. Second, the large variability among images of the same familiar face identity also minimizes the reliance on specific local features, making it difficult to recognize all or a large proportion of the famous face identities appearing upside-down.

Despite the decrease of response to inverted faces, the response was not completely abolished in this orientation, in particular for some of the individual faces providing the largest familiar individual face recognition response (e.g. Gerard Depardieu; figure 4). Precisely, the magnitude of the neural response varied as a function of face identity, with some of the French celebrities eliciting much larger responses than others. This finding probably reflects differences in visual discriminability, in other words, the ease of recognition of a particular familiar face in a crowd of unfamiliar faces. Discriminability of an individual face is strongly influenced by a variety of factors such as *visual familiarity* (e.g. frequency and/or recency of exposure on social media, films etc.), but also *within-person variability* across images (high or low) and a *face's distinctiveness* (how much does a face differ from an 'average' face in the population or a particular set of faces). Here, we selected the famous faces based on their level of famousness for the participants and included different male distractors relatively well matched for age and view variability. However, independently of their level of fame, it can be argued that some of these celebrity faces are physically more distinctive than others, in particular *Depardieu* and *Boon*, who had a larger head width and image width in the images than the average (i.e. preserving aspect ratio, see methods). Importantly, however, these factors were controlled by face inversion. The lack of significant interaction between inversion and famous face identity suggests that at least part of the perceptual distinctiveness of some of the faces could be reduced when images were turned upside-down. Moreover, to ensure that our participants were familiar with our six chosen celebrities, we administered familiarity questionnaires for which recognition was near ceiling for both image and name recognition. However, we acknowledge that these measures do not capture the complex interplay of factors that underlie *familiarity* or *famousness* beyond visual exposure ('having seen a face many times') and visualization based on a person's name ('I can visualise this person's face').

The individual face recognition response was maximal over occipito-temporal sites at the group level (figures 3–5) and for most individual participants (figure 6). This neural localization is in agreement with scalp localization of face-selective electrophysiological responses obtained either in standard transient stimulation paradigms (ERPs, e.g. [83]; see [84]) or as measured with FPVS (e.g. [61,65]), as well as responses to unfamiliar face discrimination (ERPs: [79,80,85]; periodic stimulation: [60,82]). This scalp topography suggests that these responses originate essentially from regions of the lateral occipital and ventral occipito-temporal cortices involved specifically in processing faces [2–5,7–9]. However, contrary to previous FPVS-EEG measures, in particular those reflecting unfamiliar face discrimination [59,60,64,69,82], there was no right hemisphere advantage here at the group level, with some individuals showing right lateralization, left lateralization, or no clear lateralization pattern (figure 6). The lack of right hemispheric lateralization could be due to several factors but we suggest that an important factor is the recording of a response reflecting a direct *contrast* between familiar and unfamiliar faces: if unfamiliar faces preferentially recruit the right hemisphere at the group level, as

also indicated by the 6 Hz response (figure 2), a right-lateralized response for individual recognition of familiar faces expressed in contrast to unfamiliar faces may no longer appear as being right lateralized. Another potential factor at play is that the neural response recorded here goes beyond 'visual familiarity' and may also reflect automatic triggering of semantic information and a famous name associated with the rapidly repeated famous identity, with the left hemisphere playing a more important role than the right hemisphere in these processes (e.g. [86,87]).

## 6.1. Potential applications

A sensitive and objective paradigm to measure the fast automatic high-level recognition of a familiar face identity among unfamiliar faces in humans can have a wide range of applications. First and foremost, it can be used to better characterize the human IFR function, e.g. how it is affected by stimulus size (i.e. what is the distance at which we can recognize a famous face identity?), contrast, occlusion, distortion, colour information, etc. Unfortunately, these issues are far from being resolved due to the lack of diagnostic measures of this function. For instance, while behavioural studies have suggested that highly familiar faces can be recognized despite large amounts of spatial distortions and/or blurring ([52,88–90]; see also [91]), these studies relied on single—sometimes highly iconic—stimuli presented for each celebrity under low or unlimited time constraints and without requiring discrimination of concurrently presented unfamiliar face distractors. It is unlikely that the more ecologically valid measure used here would not be affected substantially by manipulations such as distortion, blurring or low contrast levels. Second, the paradigm is suitable to inform about the neural basis of human familiar face recognition [6,92] either with human intracerebral recordings offering a high temporal resolution to measure fast frequency responses ([93]; see [94]) or adapted to metabolic neuroimaging measures as in functional magnetic resonance imaging (fMRI) [9]. Third, due to its simplicity and the lack of an explicit task, the paradigm could also be used as such to test children or even infants to inform about the development of IFR without confounding factors (i.e. developmental differences in cognitive processes involved in explicit tasks; see [95]). It is also applicable in clinical population, e.g. patients with prosopagnosia, semantic dementia, Alzheimer's disease or other neurological or psychiatric disorders with IFR impairments, who might have difficulties in task understanding and/or decisional processes in explicit recognition tasks. Finally, such a paradigm may prove invaluable for forensic research allowing to determine whether or not someone is familiar with a given individual in an implicit and very fast manner.

## 6.2. Limitations and extensions

With respect to the last point, an important issue is the extent to which the neural response in this paradigm is sensitive at the individual level. Impressively, despite a relatively short recording time, the IFR response was significant in 14 out of 16 participants. For reference, this could be compared to a recent ERP study in which, despite a considerably longer recording time, reliable differences between natural images of a celebrity and unfamiliar faces were detected in only 5 of 18 participants [41]. However, in our study, the remaining two participants in which no significant response was recorded were also highly familiar with the faces, as assessed with questionnaires, suggesting that there is room for improvement of the measure. This could be achieved by averaging more stimulation sequences to increase SNR or optimizing stimulation parameters, in particular the visual stimulation frequency. Here we used 6 Hz as the generic driving frequency because the largest neural responses to different unfamiliar face stimuli are recorded at this frequency rate [64] on the scalp in EEG and across the posterior cortical face network in fMRI [96]. Moreover, this face presentation rate allows only one fixation per face, which should be largely sufficient to individuate it. In fact, there is recent behavioural evidence that famous faces presented in a train of unfamiliar faces can be detected with an accuracy rate of 75% [73]. Nevertheless, further studies could sample a wider frequency range to optimize the neural individual face recognition response.

Besides sensitivity at the individual level, an important issue is whether the EEG response can be used to inform about interindividual differences in IFR ability, a particularly active area of research nowadays [97]. At this level, it is important to acknowledge that the amplitude of the IFR response at 0.86 Hz and harmonics will not only be influenced by the size of the neuronal population involved and its level of activity—which should directly relate to the targeted function—but also by general physiological factors such as skull thickness, the conductivity of tissue in between generators and captors and cortical folding affecting the orientation of the projection of the response to the scalp

[98–100]. Hence, as for unfamiliar face discrimination (see [101]), it is unlikely that variations of EEG amplitude at the tagged frequency rate across individuals of the same population will directly reflect the relative efficiency of their IFR function. Addressing this issue in future research may require the use of normalization measures or stimulus manipulations to increase functional variability across individuals. Finally, an important issue to resolve in future studies is the extent to which the neural response identified here reflects the degree of familiarity with the faces (within a given individual or at the group level). Interestingly, the magnitude of the response was the highest for French actor *Gerard Depardieu*, who was also named by participants as the easiest to recognize and visualize. Likewise, the smallest neural response was found for the least well-known celebrity among the set (i.e. *Vincent Cassel*). This suggests that the response would be significantly lower, or perhaps even absent, for a repeated unfamiliar face identity inserted among the rapidly presented series of unfamiliar faces, in line with behavioural observations that the generalization across widely variable images of an unfamiliar face identity presented among distractors is weak [29]. This should be tested in future studies by directly comparing the response to the exact same stimulus sequences in participants who are either familiar or unfamiliar with the faces. In the same vein, responses might be significantly weaker if a familiar face identity is inserted within a sequence of familiar faces only, given that individual familiar faces may share the same neural network. This would require the development or adjustment of the present approach (e.g. in combination with adaptation to various images of the same face) to obtain significant measures.

Concluding, incidentally spotting a familiar face in a crowd of strangers is a task most humans perform with ease in everyday life. Here, we introduce a novel neural measure of this individual face recognition function, combining EEG and FPVS in a paradigm that uses ecologically valid (i.e. highly variable) natural face stimuli. We show that the human brain can discriminate the face of a well-known celebrity among a periodically presented stream of unknown faces (i.e. strangers) despite high within-person image variability and fast stimulus presentation. Our findings of a biological marker for fast and automatic individual face recognition open an exciting future direction for a better understanding of IFR, its development and pathology and its application in forensic settings.

Ethics. This study was approved by the University of Louvain Biomedical Ethics Committee (ref. no. B403201111965). All participants gave their written informed consent prior to testing.

Data accessibility. The dataset in EEG time domain has been uploaded at Open Science Framework (https://osf.io/q5atd/).

Authors' contributions. F.G.S.Z. and B.R. participated in the conception of the research question and study design. F.G.S.Z. collected the data. F.G.S.Z. and X.Y. carried out the data analysis. F.G.S.Z. drafted the initial manuscript. All authors contributed to the interpretation of the results and manuscript revisions. F.G.S.Z. and X.Y. contributed equally to this manuscript. All authors gave final approval for publication.

Competing interests. We have no competing interests.

Funding. This study was supported in part by Fonds National de Recherche Scientifique awarded to B.R., and in part by a co-funded initiative by the University of Louvain and the Marie Curie Actions of the European Commission award to X.Y. (grant no. F211800013).

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
