## [Reviewer comments · Royal Society Open Science]

Review History

RSOS-181904.R0 (Original submission)

Review form: Reviewer 1

Is the manuscript scientifically sound in its present form?

Yes

Are the interpretations and conclusions justified by the results?

Yes

Is the language acceptable?

Yes

Is it clear how to access all supporting data?

Yes

Do you have any ethical concerns with this paper?

No

Have you any concerns about statistical analyses in this paper?

No

Recommendation?

Accept with minor revision (please list in comments)

Comments to the Author(s)

The study investigates the effectiveness of an EEG-based marker for automatic familiar face recognition with the aid of fast periodic visual stimulation (FPVS). The study builds upon the extensive work of the authors using FPVS methodology and confirms its applicability to the study of familiar face perception. Specifically, the authors find a substantial increase in the signal-to-noise ratio (SNR) of the frequency associated with famous face stimulus presentations. This effect is diminished but not eliminated for inverted faces in agreement with the well-known decrement in recognition performance associated with such stimuli. Overall, this is a straightforward and informative application of FPVS. The manuscript is clearly written and the implications are well outlined. I only have a couple of questions and suggestions targeting aspects of the results that may benefit from further consideration.

1. Signals associated with familiar face recognition were maximized, as expected, in occipitotemporal areas. In contrast, the general visual response to faces was maximized in a medial occipital area. The latter result seems somewhat surprising. Please discuss.
2. Relevant harmonics were selected based on the amplitude of their z values (p. 16). This is fine for an exploratory analysis but further analyses should best use an independent selection criterion to avoid circularity.
3. Please discuss the particular pattern of results occurring across different harmonics: is there any significance associated with the maximization of the SNR at 3.42Hz?
4. Regarding the design, generic stimulus presentation at 6Hz is selected based on previous work documenting its effectiveness. I wonder whether a similar rationale exists for presenting familiar faces at 0.86 HZ.
5. To better assess the effectiveness of the method, future work should consider presenting the same unfamiliar facial identity at the same frequency (0.86Hz) in separate sequences as a more stringent control. If distinctiveness is partly responsible for the current results, as the authors suggest, that may be able to drive FPVS responses even for unfamiliar faces (and, thus, lead to the mislabeling of distinctive unfamiliar faces as familiar in the case of practical applications).

A smaller point: please label the x axis of the bar graphs in Fig 5.

Review form: Reviewer 2 (Amanda Robinson)**Is the manuscript scientifically sound in its present form?**

Yes

Are the interpretations and conclusions justified by the results?

No

Is the language acceptable?

Yes

Is it clear how to access all supporting data?

Yes

Do you have any ethical concerns with this paper?

No

Have you any concerns about statistical analyses in this paper?

No

Recommendation?

Major revision is needed (please make suggestions in comments)

Comments to the Author(s)

In this paper, the authors investigated familiar face recognition using an FPVS oddball paradigm with EEG. Specifically, they showed that periodic presentation of a familiar identity amongst unfamiliar faces elicits a distinct neural response. This study investigates a really interesting topic; namely, identifying a marker of familiar face recognition. The manuscript is quite well written and the findings are interesting. However, I have some questions and concerns about the conclusions that leave me hesitant to recommend publication in its current form.

Major

- The biggest issue I have with the current study is that it does not tease apart any potential differences between a periodic response to a repeated familiar individual, and a periodic response to a repeated unfamiliar individual. Couldn't a repeated unfamiliar identity presented at 0.86 Hz also produce a 0.86 Hz response, even if the images were not identical? That is, could the results be indicative of a repeated versus non-repeated identity effect rather than a familiar versus unfamiliar effect? The inverted condition addresses this somewhat, but given the well documented face inversion effect even for unfamiliar faces, it seems likely to me that face-specific pattern recognition processes might show repetitive responses to an upright but unfamiliar identity, even though this effect is gone for inverted faces. Ideally, I would like to see a follow up study that uses the same paradigm but with a repeated unfamiliar identity. If the oddball response to the unfamiliar repeated faces are much lower than familiar repeated faces, I would be much more convinced that the current paradigm can be used for IFR purposes.
- I find it curious that all harmonics were analysed; in one part, the authors state that only significant harmonics were selected for further analysis, but it also says the harmonics were summed. Was it only the significant harmonics that were summed? There is also the matter of inclusion of possible intermodulation components. 5.14Hz and 6.85Hz (for example) might signify nonlinear integration between the base frequency (6Hz) and the oddball response (0.86Hz). Doesn't this make it difficult to interpret the harmonic sum as one process? In Figure 2 these potential intermodulation frequencies appear to be right lateralised, whereas the other frequencies are left lateralised. This might indeed indicate they are markers of different (interesting!) processes.
- It seems that the inverted familiarity response at 0.86Hz has a different topography than that of the upright faces. Thus, while it appears in Figures 2 and 3 that there is no oddball response for inverted faces over the lateral posterior ROIs, it might be significant over central electrodes. Could the authors touch on this by elaborating in the results and discussion?
- In the discussion, the authors state that the magnitude of the neural response varied as a function of face identity. In the results, they state that there was a main effect of face identity, but do not report simple effects tests to show differences across the identities. These should be stated. The discussion mentions that the most familiar individual has the largest oddball response, but not statistics were performed to support this statement. Is there a statistical relationship between the pre-experimental identity familiarity and the neural responses for each identity? This would be more convincing that the oddball response for upright faces is familiarity specific.

- It seems that the general face response at 6Hz was maximal at central occipital electrodes. Is there a reason why the standard right-lateralised signal was not observed?
- In the methods, it says that image height was the same, but width varied across different images. Did image width vary significantly across identities? For example, is it possible that Nicolas Sarkozy images were narrower than the unfamiliar images?
- The authors mention multiple times throughout the manuscript that a 6Hz frequency allows only one gaze fixation on each face. Is there a citation for this? Weren't participants instructed to fixate centrally and indeed perform a task at fixation?
- The use of the term "posterior ROI" is unclear to me on pages 21-22. Is this the mean of the left and right occipitotemporal electrodes as in Figs 2 and 3?

Minor

- P15: "for each participant, we collapsed individual stimulation sequences across the six famous identities in the time domain". Does this mean that the neural responses were averaged in the time domain?
- Perhaps it is just me, but the use of the term "individual familiar face recognition" seems a bit ambiguous. For instance, the individual could refer to the face being looked at, or the person viewing the face. That is, the term could imply individual differences in participants rather than a high vs low familiarity effect.
- P1, Line 12: "neurotypical human adults can spot a familiar face... without being able to suppress familiarity recognition" - is there a reference for this?
- P2, Line 33: description of prosopagnosia as "spectacular" seems a bit too positive and outrageous. Perhaps it is worth toning down this language.
- Figure 1: might be worth mentioning the task in the figure or legend.
- P16, line 54: "significant" should be "significance".

Decision letter (RSOS-181904.R0)

11-Feb-2019

Dear Dr Rossion,

The editors assigned to your paper ("An objective and sensitive neural measure of human familiar individual face recognition") have now received comments from reviewers. We would like you to revise your paper in accordance with the referee and Associate Editor suggestions which can be found below (not including confidential reports to the Editor). Please note this decision does not guarantee eventual acceptance.

Please submit a copy of your revised paper before 06-Mar-2019. Please note that the revision deadline will expire at 00.00am on this date. If we do not hear from you within this time then it will be assumed that the paper has been withdrawn. In exceptional circumstances, extensions may be possible if agreed with the Editorial Office in advance. We do not allow multiple rounds of revision so we urge you to make every effort to fully address all of the comments at this stage. If deemed necessary by the Editors, your manuscript will be sent back to one or more of the original reviewers for assessment. If the original reviewers are not available, we may invite new reviewers.

To revise your manuscript, log into <http://mc.manuscriptcentral.com/rsos> and enter your Author Centre, where you will find your manuscript title listed under "Manuscripts with Decisions." Under "Actions," click on "Create a Revision." Your manuscript number has been

appended to denote a revision. Revise your manuscript and upload a new version through your Author Centre.

- Data accessibility

If you wish to submit your supporting data or code to Dryad (<http://datadryad.org/>), or modify your current submission to dryad, please use the following link:
<http://datadryad.org/submit?journalID=RSOS&manu=RSOS-181904>

- Competing interests

- Authors' contributions

- Acknowledgements

- Funding statement

on behalf of Dr Isabelle Mareschal (Associate Editor) and Essi Viding (Subject Editor)
 openscience@royalsociety.org

Associate Editor's comments (Dr Isabelle Mareschal):

Associate Editor: 1

Comments to the Author:

Reviewers have read your paper and raised some important issues that would need to be addressed, particularly the issue of repeated vs non-repeated identity effects rather than familiar vs non familiar effects. Please provide a point by point reply to all their queries.

Comments to Author:

Reviewers' Comments to Author:

Reviewer: 1

Comments to the Author(s)

The study investigates the effectiveness of an EEG-based marker for automatic familiar face recognition with the aid of fast periodic visual stimulation (FPVS). The study builds upon the extensive work of the authors using FPVS methodology and confirms its applicability to the study of familiar face perception. Specifically, the authors find a substantial increase in the signal-to-noise ratio (SNR) of the frequency associated with famous face stimulus presentations. This effect is diminished but not eliminated for inverted faces in agreement with the well-known decrement in recognition performance associated with such stimuli. Overall, this is a straightforward and informative application of FPVS. The manuscript is clearly written and the implications are well outlined. I only have a couple of questions and suggestions targeting aspects of the results that may benefit from further consideration.

1. Signals associated with familiar face recognition were maximized, as expected, in occipitotemporal areas. In contrast, the general visual response to faces was maximized in a medial occipital area. The latter result seems somewhat surprising. Please discuss.
2. Relevant harmonics were selected based on the amplitude of their z values (p. 16). This is fine for an exploratory analysis but further analyses should best use an independent selection criterion to avoid circularity.

3. Please discuss the particular pattern of results occurring across different harmonics: is there any significance associated with the maximization of the SNR at 3.42Hz?
4. Regarding the design, generic stimulus presentation at 6Hz is selected based on previous work documenting its effectiveness. I wonder whether a similar rationale exists for presenting familiar faces at 0.86 Hz.
5. To better assess the effectiveness of the method, future work should consider presenting the same unfamiliar facial identity at the same frequency (0.86Hz) in separate sequences as a more stringent control. If distinctiveness is partly responsible for the current results, as the authors suggest, that may be able to drive FPVS responses even for unfamiliar faces (and, thus, lead to the mislabeling of distinctive unfamiliar faces as familiar in the case of practical applications).

A smaller point: please label the x axis of the bar graphs in Fig 5.

Reviewer: 2

Comments to the Author(s)

In this paper, the authors investigated familiar face recognition using an FPVS oddball paradigm with EEG. Specifically, they showed that periodic presentation of a familiar identity amongst unfamiliar faces elicits a distinct neural response. This study investigates a really interesting topic; namely, identifying a marker of familiar face recognition. The manuscript is quite well written and the findings are interesting. However, I have some questions and concerns about the conclusions that leave me hesitant to recommend publication in its current form.

Major

- The biggest issue I have with the current study is that it does not tease apart any potential differences between a periodic response to a repeated familiar individual, and a periodic response to a repeated unfamiliar individual. Couldn't a repeated unfamiliar identity presented at 0.86 Hz also produce a 0.86 Hz response, even if the images were not identical? That is, could the results be indicative of a repeated versus non-repeated identity effect rather than a familiar versus unfamiliar effect? The inverted condition addresses this somewhat, but given the well documented face inversion effect even for unfamiliar faces, it seems likely to me that face-specific pattern recognition processes might show repetitive responses to an upright but unfamiliar identity, even though this effect is gone for inverted faces. Ideally, I would like to see a follow up study that uses the same paradigm but with a repeated unfamiliar identity. If the oddball response to the unfamiliar repeated faces are much lower than familiar repeated faces, I would be much more convinced that the current paradigm can be used for IFR purposes.
- I find it curious that all harmonics were analysed; in one part, the authors state that only significant harmonics were selected for further analysis, but it also says the harmonics were summed. Was it only the significant harmonics that were summed? There is also the matter of inclusion of possible intermodulation components. 5.14Hz and 6.85Hz (for example) might signify nonlinear integration between the base frequency (6Hz) and the oddball response (0.86Hz). Doesn't this make it difficult to interpret the harmonic sum as one process? In Figure 2 these potential intermodulation frequencies appear to be right lateralised, whereas the other frequencies are left lateralised. This might indeed indicate they are markers of different (interesting!) processes.
- It seems that the inverted familiarity response at 0.86Hz has a different topography than that of the upright faces. Thus, while it appears in Figures 2 and 3 that there is no oddball response for inverted faces over the lateral posterior ROIs, it might be significant over central electrodes. Could the authors could touch on this by elaborating in the results and discussion?

- In the discussion, the authors state that the magnitude of the neural response varied as a function of face identity. In the results, they state that there was a main effect of face identity, but do not report simple effects tests to show differences across the identities. These should be stated. The discussion mentions that the most familiar individual has the largest oddball response, but not statistics were performed to support this statement. Is there a statistical relationship between the pre-experimental identity familiarity and the neural responses for each identity? This would be more convincing that the oddball response for upright faces is familiarity specific.
- It seems that the general face response at 6Hz was maximal at central occipital electrodes. Is there a reason why the standard right-lateralised signal was not observed?
- In the methods, it says that image height was the same, but width varied across different images. Did image width vary significantly across identities? For example, is it possible that Nicolas Sarkozy images were narrower than the unfamiliar images?
- The authors mention multiple times throughout the manuscript that a 6Hz frequency allows only one gaze fixation on each face. Is there a citation for this? Weren't participants instructed to fixate centrally and indeed perform a task at fixation?
- The use of the term "posterior ROI" is unclear to me on pages 21-22. Is this the mean of the left and right occipitotemporal electrodes as in Figs 2 and 3?

Minor

- P15: "for each participant, we collapsed individual stimulation sequences across the six famous identities in the time domain". Does this mean that the neural responses were averaged in the time domain?
- Perhaps it is just me, but the use of the term "individual familiar face recognition" seems a bit ambiguous. For instance, the individual could refer to the face being looked at, or the person viewing the face. That is, the term could imply individual differences in participants rather than a high vs low familiarity effect.
- P1, Line 12: "neurotypical human adults can spot a familiar face... without being able to suppress familiarity recognition" - is there a reference for this?
- P2, Line 33: description of prosopagnosia as "spectacular" seems a bit too positive and outrageous. Perhaps it is worth toning down this language.
- Figure 1: might be worth mentioning the task in the figure or legend.
- P16, line 54: "significant" should be "significance".

Author's Response to Decision Letter for (RSOS-181904.R0)

See Appendix A.

RSOS-181904.R1 (Revision)

Review form: Reviewer 1

Is the manuscript scientifically sound in its present form?

Yes

Are the interpretations and conclusions justified by the results?

Yes

Is the language acceptable?

Yes

Is it clear how to access all supporting data?

Yes

Do you have any ethical concerns with this paper?

No

Have you any concerns about statistical analyses in this paper?

No

Recommendation?

Accept as is

Comments to the Author(s)

The authors have addressed my concerns.

Decision letter (RSOS-181904.R1)

10-May-2019

Dear Dr Rossion,

I am pleased to inform you that your manuscript entitled "An objective, sensitive and ecologically valid neural measure of rapid human individual face recognition" is now accepted for publication in Royal Society Open Science.

Kind regards,

Andrew Dunn

on behalf of Dr Isabelle Mareschal (Associate Editor) and Essi Viding (Subject Editor)

Reviewer comments to Author:

Reviewer: 1

Comments to the Author(s)

The authors have addressed my concerns.

Appendix A

Replies to comments from the editors and reviewers:

-Reviewer 1

1. Signals associated with familiar face recognition were maximized, as expected, in occipitotemporal areas. In contrast, the general visual response to faces was maximized in a medial occipital area. The latter result seems somewhat surprising. Please discuss.

Reply: It is correct and a good point to raise, although it is not so surprising when one considers the factors driving the response, and how it is computed. When using natural images, the response at 6Hz and harmonics reflects the response of the visual system to face stimuli that change in low-level properties and general high-level properties (i.e., shape changes) in addition to a change of face identity 6 times by second. Moreover, with sinusoidal contrast stimulation as we used here, faces follow a uniform grey background 6 times by second. Hence, there are changes of low-level visual properties at this rate also, not just high-level visual properties (i.e., changes of facial identities). Note that despite this, if we consider only the response at 6 Hz (not the harmonics), its maximal distribution is both on medial occipital sites and the right occipito-temporal areas (see illustration below), as in the study of Alonso-Prieto et al., 2013 (with different unfamiliar faces presented at 6 Hz, and the response measured at 6 Hz, see Fig.3 in that paper).

However, the harmonics (12 Hz, 18 Hz, etc.) are all associated with a medial occipital topography because these responses are too fast to be directly related to the changes of identity (see Alonso-Prieto et al., 20103). That is, they reflect mainly low-level processes (e.g., populations of neurons discharging similarly to face onset and face offset, i.e., 12 times/second) (see Figure below).

Thus, when calculating a global response at the base rate (6 Hz + 12 Hz + 18 Hz, etc.), it is dominated by the low-level contribution. Similar response patterns have also been observed at the base rate in our previous studies with the same FPVS approach measuring generic face categorization (e.g., Retter & Rossion, 2016, Neuropsychologia; Quek et al., 2018, JOCN, see Figure 3 in that study, comparing 6 Hz and 12 Hz) or unfamiliar individual face discrimination (Liu-Shuang et al., 2014, 2016; Dzhelyova & Rossion, 2014a, 2014b).

Yet, it is very interesting because the spread of activation to the right occipito-temporal cortex is not only almost limited to the first harmonic (6 Hz) but to upright faces.

Therefore, we have added a figure (page 20, the new Figure 2, see below also) illustrating amplitude spectra of general visual responses at the base rate with 3D topographies for each significant harmonic, an analysis of the response also separately for the first harmonics, and a brief discussion in the method part (page 13) and discussion part (page 26-27). We thank the reviewer especially for raising this point because even though it's not the main goal of the study, we believe that this modification enriches the manuscript.

2. Relevant harmonics were selected based on the amplitude of their z values (p. 16). This is fine for an exploratory analysis but further analyses should best use an independent selection criterion to avoid circularity.

Reply: We use this approach to define the harmonics, which are then quantified. The statistics used to define the harmonics (Z-scores) is not the statistics included in the subsequent analysis. Moreover, the harmonics are defined based on a combination across the two orientation conditions, all channels and all subjects, precisely to avoid any selection bias. Then, we quantify the response separately for each condition, subject and channel. Given that here it is a new experiment, we cannot use previous knowledge to define the number of harmonics to include for instance. An alternative would be to use one half of the data to define the harmonics and the other half to quantify the response, but this would result in the loss of half of the data, while, in reality, the specific approach used does not change at all the conclusions. Indeed, as what we found in our previous face perception studies with the same FPVS approach, harmonic response amplitude for the discrimination response generally decreases significantly beyond 6 Hz. In the current experiment, starting from the 10th harmonic (i.e. 8.6Hz), the amplitude of higher harmonics reduced dramatically to nearly 0 μ V.

However, to take into account this point, we did a further ANOVA (Orientation \times Hemisphere) on the recognition responses by summing up the first 18 harmonics of interest frequency (i.e. up to 17.14 Hz, excluding the 7th and the 14th harmonics that were overlapping with the base frequency). We found consistent results compared to our original analysis in which only the first 9 significant harmonics (skipping the 6Hz harmonic that was overlapping with the stimulus presentation rate, $z > 3.1$, $p < .001$, one-tailed) were considered. There was a significant main effect of Orientation ($F(1,15) = 19.54$, $p < .001$). Neither the main effect of Hemisphere ($F(1,15) = 2.5$, $p = .14$) or the interaction of Hemisphere \times Orientation ($F(1,15) = 0.4$, $p = .5$) was significant. We have clarified in the methods section (page 16) the selection of harmonics, and we are confident that any type of reasonable selection of harmonics will lead to the same conclusion (our EEG data segmented in the time-domain will be available for anybody to replicate our results).

3. Please discuss the particular pattern of results occurring across different harmonics: is there any significance associated with the maximization of the SNR at 3.42Hz?

Reply: Note that the original figure 2 (page 19) – which only serves for illustrative purposes - has been revised because it was based initially on a SNR transformation after averaging across individual participants in the time domain. We have corrected that in the paper (now Figure 3 in the revised manuscript, page 21). The detailed analysis steps has also been added in the method part, please see page 16. The analysis shows a slightly different response patterns across harmonics to the recognition of individual upright familiar faces (with the highest SNR at 4.28 Hz), and no significant harmonics were detected when faces were all upside-down. Also, it shows SNR values for visualization, but the relative SNR values do not necessarily translate into larger amplitudes (e.g., harmonics associated with a large amplitude but in a low frequency range around 1 Hz or in the alpha band will have their SNR reduced because of the presence of noise. Please see below for a comparison of SNR and baseline corrected amplitudes. While the SNR is the highest at 4.28 Hz, this is due to a decrease of noise more than an increase of signal, as shown on the figure in the right.

As has been discussed in detail in a previous study (Retter & Rossion, 2016), the selective response in such paradigms should be measured by a summation of the baseline-corrected harmonic response amplitudes. It is very difficult to attribute a specific meaning to a given harmonic because harmonics reflect the distortion of the response at 0.86 Hz. We have added a note on this issue of harmonics in the revised version (page 16).

4. Regarding the design, generic stimulus presentation at 6Hz is selected based on previous work documenting its effectiveness. I wonder whether a similar rationale exists for presenting familiar faces at 0.86 Hz.

Reply: In truth, the 6 Hz stimulation rate is based on two considerations: it provides a large response, but it also constrains the system to one fixation/face (i.e., 166 ms stimulus presentation). As for the 0.86 Hz, of course, it is not mandatory and other parameters could be used. However, some amount of time intervals between familiar faces is necessary to avoid response interferences. Again, the study of Retter & Rossion (2016) has shown that a stimulus-onset-asynchronies (SOAs) from 400 ms onward is critical to elicit stable face-selective responses among other non-face objects in a dynamic visual stream. When face stimuli were presented with a shorter SOA, there was not enough time for the response deflections to return back to baseline that the

responses in waveforms were overlapping. The overlapping might decrease the relative response amplitudes and also the latency of the deflections. In addition, traditional EEG studies have shown that a familiar face based recognition response are actually reflected by multiple ERP components, such as an occipito-temporal lateralized N250 typically reaching maximum between 230 and 280 ms after stimulus onset, but also two broadly distributed, even later evoked components, the N400 and P600, with sometimes prolonged effects until 800-1000ms (e.g., Eimer et al., 2000a; Bentin & Deouell, 2000; Gosling & Eimer, 2011; Kaufmann et al., 2008; Tanaka et al., 2006; Wiese et al., 2019). Therefore, in our current study, a relative long-time interval between two familiar face images was used (i.e., after 1s of 6 images) to ensure that we could observe a reliable individual familiar face recognition response. We believe that the time interval could have been reduced, maybe using 1.2 Hz, but we were conservative. We briefly justify the use of 0.86 Hz in the methods section of the revised version (page 13): “At 0.86 Hz, the temporal distance between two images of the familiar face identity was 1167ms, ensuring that specific responses to familiar faces would not overlap with one another.”

5. To better assess the effectiveness of the method, future work should consider presenting the same unfamiliar facial identity at the same frequency (0.86Hz) in separate sequences as a more stringent control. If distinctiveness is partly responsible for the current results, as the authors suggest, that may be able to drive FPVS responses even for unfamiliar faces (and, thus, lead to the mislabeling of distinctive unfamiliar faces as familiar in the case of practical applications).

Reply: *This is completely true. Our goal here was to show that the paradigm works for reliably discriminating familiar from unfamiliar faces, as we see it as the key process to measure (as presented in the introduction). We also control for a contribution of low-level visual cues by including the condition with inverted faces. We have been working on an extension of this work with unfamiliar faces. However, we realized that to clearly demonstrate how much of the effect is due to familiarity and avoid image-based confounds, one needs to use the exact same image sequences, comparing people who are familiar vs. unfamiliar with the face identities. Thus, this requires testing subjects abroad because we used the most familiar faces for the local population tested. We discuss this issue – also raised by reviewer 2 – in the revised version (page 35, “This suggests that the response ...”).*

A smaller point: please label the x axis of the bar graphs in Fig 5.

Reply: *Figure 5 is showing each participants’ mean summed-harmonic familiar face recognition response in baseline-corrected amplitude (μV). Each subplot is showing the amplitudes of 45 frequency bins with the middle one indicating the frequency bin of our interest (e.g., 0.86 Hz). The unit of the x-axis of each subplot is arbitrary. We have changed the format of the Figure 5 to show a better response patterns in each individual.*

We thank the reviewer for their careful reading and constructive criticisms of our manuscript.

-Reviewer 2

Major

- The biggest issue I have with the current study is that it does not tease apart any potential differences between a periodic response to a repeated familiar individual, and a periodic response to a repeated unfamiliar individual. Couldn't a repeated unfamiliar identity presented at 0.86 Hz also produce a 0.86 Hz response, even if the images were not identical? That is, could the results be indicative of a repeated versus non-repeated identity effect rather than a familiar versus unfamiliar effect? The inverted condition addresses this somewhat, but given the well documented face inversion effect even for unfamiliar faces, it seems likely to me that face-specific pattern recognition processes might show repetitive responses to an upright but unfamiliar identity, even though this effect is gone for inverted faces. Ideally, I would like to see a follow up study that uses the same paradigm but with a repeated unfamiliar identity. If the oddball response to the unfamiliar repeated faces are much lower than familiar repeated faces, I would be much more convinced that the current paradigm can be used for IFR purposes.

Reply: This is a good point, also raised by Reviewer 1 above, please see our response above, and the discussion in the manuscript.

- I find it curious that all harmonics were analysed; in one part, the authors state that only significant harmonics were selected for further analysis, but it also says the harmonics were summed. Was it only the significant harmonics that were summed?

Reply: We agree that this point was a little confusing in the previous version. For the analysis, only the significant harmonics were considered indeed, as defined on grand-averaged data across conditions, and channels (as in our previous studies with this approach, see e.g., Retter & Rossion, 2016). They were then summed as a quantitative measure of the responses at our two stimulation frequencies. We have rephrased the sentences in the method part, please see page 16, "To take in to account the multiple harmonics tested, ..."

There is also the matter of inclusion of possible intermodulation components. 5.14Hz and 6.85Hz (for example) might signify nonlinear integration between the base frequency (6Hz) and the oddball response (0.86Hz). Doesn't this make it difficult to interpret the harmonic sum as one process? In Figure 2 these potential intermodulation frequencies appear to be right lateralised, whereas the other frequencies are left lateralised. This might indeed indicate they are markers of different (interesting!) processes.

Reply: It's an interesting point. But technically, every harmonic, even 0.86 Hz, could partly reflect an IM between the base rate and the "oddball" rate (if 0.86 Hz is F2 stimulation and 6 Hz is F1 stimulation, a response at 0.86 Hz could reflect not only f2 but also f1-6f2) even though this is really far-stretched (in our experience, IMs are of much smaller amplitude than fundamental driving responses, e.g., Boremanse et al., 2013, J. Vision). In any case, all of these responses would be related to the discrimination of the familiar from unfamiliar faces, and could not appear if this discrimination was not performed (hence, for inverted faces, the 6 Hz response is almost as large as for upright faces, but there is virtually no response to the familiar faces). It could be indeed that different harmonics reflect different processes, but in the presence of the current data, one could only speculate about that (for instance, in infants, contrasting faces to objects in this paradigm gives a response at only the first harmonic, de Heering & Rossion, 2015, while the response spreads over multiple harmonics in the same paradigm as adults. We have added a short note on the harmonics quantification in the revised version (page 16), but in the absence of further

evidence for modulations of specific harmonics related to well-defined experimental factors, we cannot speculate too much.

- It seems that the inverted familiarity response at 0.86Hz has a different topography than that of the upright faces. Thus, while it appears in Figures 2 and 3 that there is no oddball response for inverted faces over the lateral posterior ROIs, it might be significant over central electrodes. Could the authors could touch on this by elaborating in the results and discussion?

Reply: We have reported the results comparing the mean amplitudes over medial ROI between two face orientation conditions in the original paper. Yes, there was a significant effect, but there was no difference between these two conditions. We also updated Figure 4 to include responses over the medial occipital ROI for both orientation conditions. Please see page 23, highlighted text, and a brief discussion on page 31.

- In the discussion, the authors state that the magnitude of the neural response varied as a function of face identity. In the results, they state that there was a main effect of face identity, but do not report simple effects tests to show differences across the identities. These should be stated. The discussion mentions that the most familiar individual has the largest oddball response, but not statistics were performed to support this statement. Is there a statistical relationship between the pre-experimental identity familiarity and the neural responses for each identity? This would be more convincing that the oddball response for upright faces is familiarity specific.

Reply: This is a good point also. The simple effect analysis of the main effect of Identity has been added in the revised manuscript. Please see page 24-25. The recognition amplitude to Gerard Depardieu was significantly larger than almost all other face identities, except one (Dany Boon). The original aim of this study was to provide a robust and highly sensitive neural index of individual familiar face recognition, so we selected celebrity faces that were supposed to be highly familiar to our participants. It turned out to be that all our 16 participants could very well recognize 5 of the celebrity faces, but only 13 of them could recognize Vincent Cassel. According to the simple effect analysis, the response amplitude to Vincent Cassel was significantly lower than Gerard Depardieu and Nicolas Sarkozy. Unfortunately, we did not include the measure of participants' subjective familiarity extent to each face identity in our questionnaire to be able to investigate the relationship between the pre-experimental identity familiarity and the neural responses. This should be explored in future studies.

- It seems that the general face response at 6Hz was maximal at central occipital electrodes. Is there a reason why the standard right-lateralised signal was not observed?

Reply: This point was discussed above in response to reviewer 1, and we copy-paste the reply above:

It is correct and a good point to raise, although it is not so surprising when one considers the factors driving the response, and how it is computed. When using natural images, the response at 6Hz reflects the response of the visual system to face stimuli that change in low-level properties, general high-level properties (i.e., shape changes) in addition to a change of face identity 6 times by second. Moreover, with sinusoidal contrast stimulation as we used here, faces follow a uniform background 6 times by second. Hence, there are changes of low-level visual properties at this rate also, not just high-level visual properties (i.e., changes of facial identities). Note that despite this, if we consider only the response at 6 Hz (not the harmonics), its maximal distribution is

on the right occipito-temporal areas (see illustration below), as in the study of Alonso-Prieto et al., 2013 (with different unfamiliar faces presented at 6 Hz, and the response measured at 6 Hz, see Fig.3 in that paper).

However, the harmonics (12 Hz, 18 Hz, etc.) are all associated with a medial occipital topography because these responses are too fast to be directly related to the changes of identity (see Alonso-Prieto et al., 2010), and they reflect mainly low-level processes (e.g., populations of neurons discharging similarly to face onset and face offset, i.e., 12 times/second) (see Figure below).

Thus, when calculating a global response at the base rate (6 Hz + 12 Hz + 18 Hz, etc.), it is dominated by the low-level contribution. Similar response patterns have also been observed at the base rate in our previous studies with the same FPVS approach measuring generic face categorization (e.g., Retter & Rossion, 2016, *Neuropsychologia*; Quek et al., 2018, *JOCN*, see Figure 3 in that study, comparing 6 Hz and 12 Hz) or unfamiliar individual face discrimination (Liu-Shuang et al., 2014, 2016; Dzhelyova & Rossion, 2014a, 2014b).

Yet, it is very interesting because the spread of activation to the right occipito-temporal cortex is not only almost limited to the first harmonic (6 Hz) but to upright faces.

Therefore, we have added a figure (page 20, the new Figure 2, see below also) illustrating amplitude spectra of general visual responses at the base rate with 3D topographies for each significant harmonic, an analysis of the response also separately for the first harmonics, and a brief discussion in the method part (page 13) and the discussion part (page 26-27). We thank the reviewer especially for raising this point because even though it's not the main goal of the study, this modification enriches the manuscript.

- In the methods, it says that image height was the same, but width varied across different images. Did image width vary significantly across identities? For example, is it possible that Nicolas Sarkozy images were narrower than the unfamiliar images?

Reply: Indeed, image width varied across identities. While there was no significant difference in image width between unfamiliar face images and 4 familiar face identities, natural images of Gerard Depardieu and Dany Boon are significantly wider than the

unfamiliar images. This is because their head size was significantly larger. Equalizing for image width would have led to artificial reduction of these two individuals head size, or they would have taken a larger space on the images on average (i.e., with less background). We chose to keep head area / image area ratios constant across the 6 familiar face identities and the unfamiliar faces. We acknowledge that there is no perfect control when using such natural images, and this is why it is important to compare upright and inverted stimuli. We have added a note on this issue in the revised version of the manuscript (page 11).

- The authors mention multiple times throughout the manuscript that a 6Hz frequency allows only one gaze fixation on each face. Is there a citation for this? Weren't participants instructed to fixate centrally and indeed perform a task at fixation?

Reply: Yes, they have to perform a task at fixation, but studies have shown that usually after the onset of a target for a saccade, it takes about 200 ms for eye movement to begin (Fischer & Rampsberger, 1984). Therefore, here we can say that a reliable and significant individual familiar face recognition response could be achieved with only a single glance.

In the method part, we have mentioned that our participants had to respond to the color change of a central fixation cross, while at the same time monitor the flickering face stimuli. Please see page 12, section PROCEDURE-Visual stimulation, highlighted text.

- The use of the term "posterior ROI" is unclear to me on pages 21-22. Is this the mean of the left and right occipitotemporal electrodes as in Figs 2 and 3?

Reply: The "posterior ROI" indeed referred to the occipito-temporal region-of-interest. For consistency and clarity, in the revised version of the manuscript we have changed it all to the "OT ROI".

Minor

- P15: "for each participant, we collapsed individual stimulation sequences across the six famous identities in the time domain". Does this mean that the neural responses were averaged in the time domain?

Reply: Yes. Before applying a Fast Fourier Transform (FFT) to the time domain signal, we first combined and averaged across the EEG signals in time domain of all six face identity conditions (upright and inverted faces, separately) to reduce EEG activity that were not phase-locked to the stimulus. A FFT was then applied to the averaged data segments to represent the data of each channel as a normalized amplitude spectrum (μV) in the frequency domain. Note that we also performed a FFT analysis separately for each identity. The methods have been clarified on this point (page 15).

- Perhaps it is just me, but the use of the term "individual familiar face recognition" seems a bit ambiguous. For instance, the individual could refer to the face being looked at, or the person viewing the face. That is, the term could imply individual differences in participants rather than a high vs low familiarity effect.

Reply: In the revised version, we use the term individual face recognition throughout (we removed "familiar"), and we define it at the beginning of the paper. Note that "individual recognition" is the term used by biologists to define the ability to recognize

specific individuals in the group (Tibbetts & Dale, 2007, Trends in Ecology and Evolution Vol.22 No.10).

- P1, Line 12: “neurotypical human adults can spot a familiar face... without being able to suppress familiarity recognition” – is there a reference for this?

Reply: It refers to our common experience although familiar face recognition and its mechanisms has been investigated extensively since the mid 1960s in the laboratory experimental environment, and the ease at detecting a familiar face has led many researchers to suggest that familiar face recognition is somehow automatic (see Palermo & Rhodes, 2007 for a review). Our results completely support that.

- P2, Line 33: description of prosopagnosia as “spectacular” seems a bit too positive and outrageous. Perhaps it is worth toning down this language.

Reply: This has been corrected indeed.

- Figure 1: might be worth mentioning the task in the figure or legend.

Reply: We have added the task in the legend of Figure 1, see page 9.

- P16, line 54: “significant” should be “significance”.

Reply: Thank you for noticing this typo. This has been corrected.

We thank the reviewer for their careful reading and constructive criticisms of our manuscript.